# Lightweight Vision Transformer with Bidirectional Interaction

**Qihang Fan** [1,2], **Huaibo Huang**[1], **Xiaoqiang Zhou**[1,3], **Ran He**[1,2*]

[1]MAIS & CRIPAC, Institute of Automation, Chinese Academy of Sciences, Beijing, China
[2]School of Artificial Intelligence, University of Chinese Academy of Sciences, Beijing, China
[3]University of Science and Technology of China, Hefei, China
fanqihang.159@gmail.com, huaibo.huang@cripac.ia.ac.cn,
xq525@mail.ustc.edu.cn, rhe@nlpr.ia.ac.cn

## Abstract

Recent advancements in vision backbones have significantly improved their performance by simultaneously modeling images' local and global contexts. However, the bidirectional interaction between these two contexts has not been well explored and exploited, which is important in the human visual system. This paper proposes a **F**ully **A**daptive **S**elf-**A**ttention (FASA) mechanism for vision transformer to model the local and global information as well as the bidirectional interaction between them in context-aware ways. Specifically, FASA employs self-modulated convolutions to adaptively extract local representation while utilizing self-attention in down-sampled space to extract global representation. Subsequently, it conducts a bidirectional adaptation process between local and global representation to model their interaction. In addition, we introduce a fine-grained downsampling strategy to enhance the down-sampled self-attention mechanism for finer-grained global perception capability. Based on FASA, we develop a family of lightweight vision backbones, **F**ully **A**daptive **T**ransformer (FAT) family. Extensive experiments on multiple vision tasks demonstrate that FAT achieves impressive performance. Notably, FAT accomplishes a **77.6%** accuracy on ImageNet-1K using only **4.5M** parameters and **0.7G** FLOPs, which surpasses the most advanced ConvNets and Transformers with similar model size and computational costs. Moreover, our model exhibits faster speed on modern GPU compared to other models.

## 1 Introduction

Vision Transformers (ViTs) have recently garnered significant attention in the computer vision community due to their exceptional ability for long-range modeling and context-aware characteristics. However, because of the quadratic complexity of self-attention in ViT [13], its computational cost is extremely high. As a result, many studies have emerged to improve ViT's computational efficiency and performance in various ways. For instance, some methods restrict tokens that perform self-attention to a specific region and introduce inductive bias to ViT [34; 12; 66; 55]. Further, some methods aim to transform ViT into lightweight backbones with fewer parameters and computational requirements [38; 40; 4; 30; 37], achieving promising results but still not matching the performance of the most advanced ConvNets [51]. How to design an excellent lightweight Vision Transformer remains a challenge.

In current state-of-the-art Vision Transformers, some either excel in creating local feature extraction modules [34; 12; 66] or employing efficient global information aggregation modules [57; 58], while others incorporate both [42; 41]. For instance, LVT [66] unfolds tokens into separate windows and

---

[*]Ran He is the corresponding author.

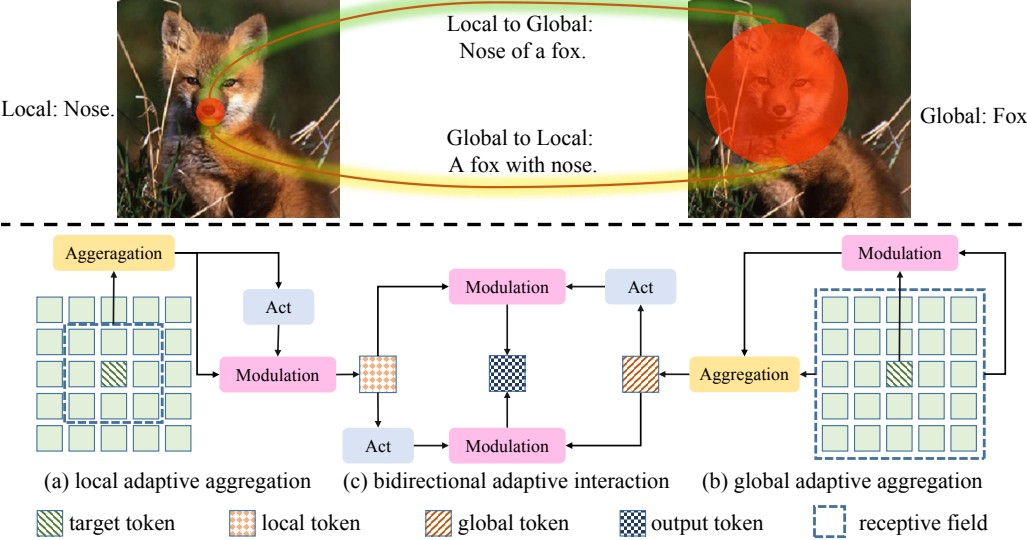

Local: Nose.

Local to Global:
Nose of a fox.

Global to Local:
A fox with nose.

Global: Fox.

(a) local adaptive aggregation   (c) bidirectional adaptive interaction   (b) global adaptive aggregation

target token   local token   global token   output token   receptive field

Figure 1: Illustration of the human visual system (top) and our FASA (bottom). The human visual system can perceive both local and global contexts and model the bidirectional interaction between them. Our FASA follows this mechanism and consists of three parts: (a) local adaptive aggregation, (b) global adaptive aggregation, and (c) bidirectional adaptive interaction. Our FASA models local information, global information, and local-global bidirectional interaction in context-aware manners.

applies self-attention within the windows to extract local features, while PVT [57; 58] leverages self-attention with downsampling to extract global features and reduce computational cost. Unlike them, LITv2 [41] relies on window self-attention and spatial reduction attention to capture local and global features, respectively. In terms of the local-global fusion, most methods use simple local-global sequential structures [38; 40; 37; 26; 82], whereas others combine local and global representation with simple linear operations through local-global parallel structures [48; 41; 43]. However, few works have investigated the bidirectional interaction between local and global information. Considering the human visual system where bidirectional local-global interaction plays an important role, these simplistic mixing methods are not fully effective in uncovering the intricate relationship between local and global contexts.

In Fig. 1, we illustrate how humans observe an object and notice its body details. Using the example of a fox, we can observe two types of interaction that occur when humans focus on either the fox's nose or the entire animal. In the first type of interaction, known as Local to Global, our understanding of the local feature transforms into the "Nose of a fox." In the second type of interaction, called Global to Local, the way we comprehend the global feature changes to the "A fox with nose." It can be seen that the bidirectional interaction between local and global features plays an essential role in the human visual system. Based on this fact, we propose that a superior visual model should not only extract good local and global features but also possess adequate modeling capabilities for their interaction.

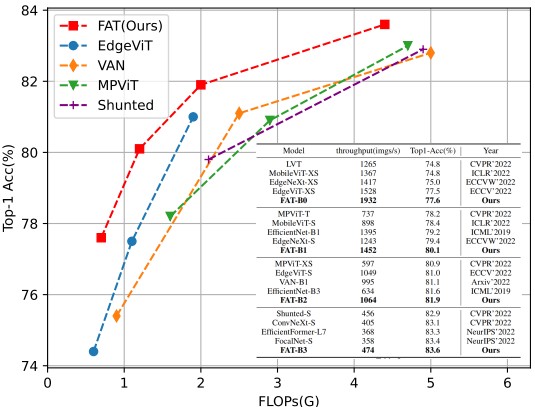

Figure 2: Top-1 accuracy v.s. FLOPs on ImageNet-1K of recent SOTA CNN and transformer models. The proposed Fully Adaptive Transformer (FAT) outperforms all the counterparts in all settings.

In this work, our objective is to model the bidirectional interaction between local and global contexts while also improving them separately. To achieve this goal, we introduce three types of Context-Aware Feature Aggregation modules. Specifically, as shown in Fig. 1, we first adaptively aggregate local and global features using context-aware manners to obtain local and global tokens, respectively. Then, we perform point-wise cross-modulation between these two types of tokens to model their

bidirectional interaction. We streamline all three processes into a simple, concise, and straightforward procedure. Since we use context-aware approaches to adaptively model all local, global, and local-global bidirectional interaction, we name our novel module the **F**ully **A**daptive **S**elf-**A**ttention (FASA). In FASA, we also further utilize a fine-grained downsampling strategy to enhance the self-attention mechanism, which results in its ability to perceive global features with finer granularity. In summary, FASA introduces only a small number of additional parameters and FLOPs, yet it significantly improves the model's performance.

Building upon FASA, we introduce the **F**ully **A**daptive **T**ransformer (FAT) family. The FATs follow the hierarchical design [34; 57] and serve as general-purpose backbones for various computer vision tasks. Through extensive experiments, including image classification, object detection, and semantic segmentation, we validate the performance superiority of the FAT family. Without extra training data or supervision, our FAT-B0 achieves a top-1 accuracy of **77.6%** on ImageNet-1K with only **4.5M** parameters and **0.7G** FLOPs, which is the first model surpasses the most advanced ConvNets with similar model size and computational cost as far as we know. Additionally, as shown in Fig. 2, our FAT-B1, B2, and B3 also achieve state-of-the-art results while maintaining similar model sizes and computational costs.

## 2 Related Work

**Vision Transformer.** Since the earliest version of ViT [13] appeared, numerous works have been proposed that focus on enhancing the self-attention mechanism. Many methods restrict self-attention to a subset of tokens, sacrificing global receptive fields to reduce the computation of ViT [34; 12; 66; 61; 1]. Despite having only local perception capabilities, the context-aware nature of self-attention enables these methods to achieve excellent performance. A famous among them is the Swin-Transformer [34], which divides all tokens into windows and performs self-attention within each window, achieving highly competitive performance. In contrast, some methods downsample the $K$ and $V$ in self-attention to preserve the global receptive field while reducing computation by minimizing the number of token pairs involved in the calculation [57; 60; 40; 47; 68; 5]. One such method, PVT [57], leverages large-stride convolution to process $K$ and $V$, effectively lowering their spatial resolution. In addition to these methods, numerous efforts have been made to introduce ViT into the design of lightweight vision backbones [26; 40; 37; 15; 38; 4]. For example, MobileViT [38] concatenates convolution and self-attention to obtain a powerful, lightweight backbone. Nevertheless, a performance gap exists between lightweight vision transformers and state-of-the-art lightweight CNNs such as NAS-based EfficientNet [51].

**Local-Global Fusion.** A high-performance vision backbone typically possesses exceptional capabilities for both local and global perception. The capabilities are achieved by either connecting local and global perception modules in a serial manner [58; 31; 10; 26; 40; 55; 1], as demonstrated by DaViT [10], or by simultaneously modeling local and global information within a single module [48; 41; 43], such as the inception transformer [48]. However, current approaches to fusing local and global information are overly simplistic. In serial models, the process of fusing local and global information is not adequately represented. In parallel structures, almost all methods rely on linear modules that depend entirely on trainable parameters to fuse local and global information [41; 48; 43]. These fusing approaches lack the ability to model the interaction between local and global information, which is inconsistent with the human visual system shown in Fig. 1. In contrast, our proposed FASA module models bidirectional interaction between local and global information while separately modeling each one.

**Self-Attention with Downsampling.** Currently, many models utilize self-attention with downsampling, a technique that was earlier used in PVT and so on to improve the computational efficiency [57; 58; 43; 41; 48; 61; 5]. PVT [57] reduces the spatial resolution of $K$ and $V$ using non-overlapping large stride convolutions, which decreases the number of token pairs involved in self-attention and lowers the computational cost while maintaining a global receptive field. Similarly, PVTv2 [58] uses large stride average pooling to downsample $K$ and $V$. In contrast, inception transformer [48] employs large stride average pooling to downsample all three $Q$, $K$, and $V$, then upsamples the tokens to their original size after self-attention is applied. However, using excessively large strides or non-overlapping downsampling as in these methods may lead to significant information loss. In FASA, we propose a fine-grained downsampling strategy to alleviate this issue.

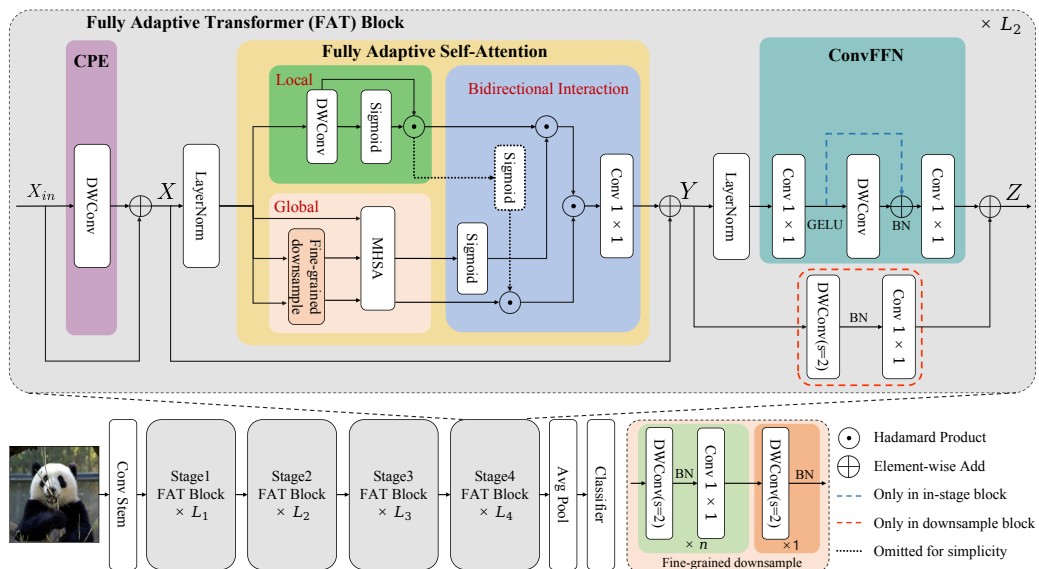

Figure 3: Illustration of the FAT. FAT is composed of multiple FAT blocks. A FAT block consists of CPE, FASA and ConvFFN.

## 3 Method

### 3.1 Overall Architecture.

The overall architecture of the Fully Adaptive Transformer (FAT) is illustrated in Fig. 3. To process an input image $x \in \mathcal{R}^{3 \times H \times W}$, we begin by feeding it into the convolutional stem used in [63]. This produces tokens of size $\frac{H}{4} \times \frac{W}{4}$. Following the hierarchical designs seen in previous works [48; 50; 46; 49], we divide FAT into four stages to obtain hierarchical representation. Then we perform average pooling on the feature map containing the richest semantic information. The obtained one-dimensional vector is subsequently classified using a linear classifier for image classification.

A FAT block comprises three key modules: Conditional Positional Encoding (CPE) [6], Fully Adaptive Self-Attention (FASA), and Convolutional Feed-Forward Network (ConvFFN). The complete FAT block is defined by the following equation (Eq. 1):

$$
\begin{aligned}
X &= \text{CPE}(X_{in}) + X_{in}, \\
Y &= \text{FASA}(\text{LN}(X)) + X, \\
Z &= \text{ConvFFN}(\text{LN}(Y)) + \text{ShortCut}(Y).
\end{aligned}
\tag{1}
$$

Initially, the input tensor $X_{in} \in \mathcal{R}^{C \times H \times W}$ passes through the CPE to introduce positional information for each token. The subsequent stage employs FASA to extract local and global representation adaptively, while a bidirectional adaptation process enables interaction between these two types of representation. Finally, ConvFFN is applied to enhance local representation further. Two kinds of ConvFFNs are employed in this process. In the in-stage FAT block, ConvFFN's convolutional stride is 1, and $\text{ShortCut} = \text{Identity}$. At the intersection of the two stages, the ConvFFN's convolutional stride is set to 2, and $\text{ShortCut} = \text{DownSample}$ is achieved via a Depth-Wise Convolution (DWConv) [23] with the stride of 2 along with a $1 \times 1$ convolution. The second type of ConvFFN accomplishes downsampling using only a small number of parameters, avoiding the necessity of patch merging modules between the two stages, thereby saving the parameters.

### 3.2 Fully Adaptive Self-Attention

In Fig. 1, we present the visual system of humans, which not only captures local and global information but also models their interaction explicitly. Taking inspiration from this, we aim to develop a similar module that can adaptively model local and global information and their interaction. This leads to the proposal of the **F**ully **A**daptive **S**elf-**A**ttention (FASA) module. Our FASA utilizes context-aware manners to model all three types of information adaptively. It comprises three modules:

global adaptive aggregation, local adaptive aggregation, and bidirectional adaptive interaction. Given the input tokens $X \in \mathcal{R}^{C \times H \times W}$, each part of the FASA will be elaborated in detail.

### 3.2.1 Definition of Context-Aware Feature Aggregation

In order to provide readers with a clear comprehension of FASA, we will begin by defining the various methods of context-aware feature aggregation (CAFA). CAFA is a widely used feature aggregation method. Instead of solely relying on shared trainable parameters, CAFA generates token-specific weights based on the target token and its local or global context. These newly generated weights, along with the associated context of the target token, are then used to modulate the target token during feature aggregation, enabling each token to adapt to its related context. Generally, CAFA consists of two processes: aggregation ($\mathcal{A}$) and modulation ($\mathcal{M}$). In the following, the target token is denoted by $x_i$, and $\mathcal{F}$ represents a non-linear activation function. Based on the order of $\mathcal{A}$ and $\mathcal{M}$, CAFA can be classified into various types. Various forms of self-attention [56; 58; 57; 45] can be expressed as Eq. 2. Aggregation over the contexts $X$ is performed after the attention scores between query and key are computed. The attention scores are obtained by modulating the query with the keys, and then applying a $\mathrm{Softmax}$ to the resulting values:

$$y_i = \mathcal{A}(\mathcal{F}(\mathcal{M}(x_i, X)), X), \tag{2}$$

In contrast to the approach outlined in Eq. 2, recent state-of-the-art ConvNets [22; 17; 67] utilize a different CAFA technique. Specifically, they employ DWConv to aggregate features, which are then used to modulate the original features. This process can be succinctly described using Eq. 3.

$$y_i = \mathcal{M}(\mathcal{A}(x_i, X), x_i), \tag{3}$$

In our FASA module, global adaptive aggregation improves upon the traditional self-attention method with fine-grained downsampling and can be mathematically represented by Eq. 2. Meanwhile, our local adaptive aggregation, which differs slightly from Eq. 3, can be expressed as Eq. 4:

$$y_i = \mathcal{M}(\mathcal{F}(\mathcal{A}(x_i, X)), \mathcal{A}(x_i, X)), \tag{4}$$

In terms of the bidirectional adaptive interaction process, it can be formulated by Eq. 5. Compared to the previous CAFA approaches, bidirectional adaptive interaction involves two feature aggregation operators ($\mathcal{A}_1$ and $\mathcal{A}_2$) that are modulated with each other:

$$\begin{aligned} y_{i2} &= \mathcal{M}(\mathcal{F}(\mathcal{A}_1(x_i, X)), \mathcal{A}_2(x_i, X)), \\ y_{i1} &= \mathcal{M}(\mathcal{F}(\mathcal{A}_2(x_i, X)), \mathcal{A}_1(x_i, X)). \end{aligned} \tag{5}$$

### 3.2.2 Global Adaptive Aggregation

The inherently context-aware nature and capacity to model long-distance dependencies of self-attention make it highly suitable for adaptively extracting global representation. As such, we utilize self-attention with downsampling for global representation extraction. In contrast to other models that downsample tokens [57; 58; 60; 48] by using large stride convolutions or pooling operations, we adopt a fine-grained downsampling strategy to minimize loss of global information to the greatest extent possible. In particular, our fine-grained downsampling module is composed of several basic units. Each unit utilizes a DWConv with a kernel size of $5 \times 5$ and a stride of 2, followed by a $1 \times 1$ convolution that subtly downsamples both $K$ and $V$. After that, $Q$, $K$, and $V$ are processed through the Multi-Head Self-Attention (MHSA) module. Unlike regular MHSA, we omit the last linear layer. The complete procedure for the global adaptive aggregation process is illustrated in Eq. 6 and Eq. 7. Fisrt, we define our fine-grained downsample strategy and the base unit of it in Eq. 6:

$$\begin{aligned} \mathrm{BaseUnit}(X) &\triangleq \mathrm{Conv}_{1\times1}(\mathrm{BN}(\mathrm{DWConv}(X))), \\ \mathrm{pool}(X) &\triangleq \mathrm{BN}(\mathrm{DWConv}(\mathrm{BaseUnit}^{(n)}(X)), \end{aligned} \tag{6}$$

where $(n)$ represents the number of base units that are concatenated and $\mathrm{pool}$ denotes our fine-grained downsample operator. Then, the MHSA is conducted as Eq. 7:

$$\begin{aligned} Q, K, V &= W_1 \otimes X, \\ X'_{global} &= \mathrm{MHSA}(Q, \mathrm{pool}(K), \mathrm{pool}(V)). \end{aligned} \tag{7}$$

where the mathematical symbol $\otimes$ denotes the operation of matrix multiplication, $W_1$ is a learnable matrix.

### 3.2.3 Local Adaptive Aggregation

In contrast to self-attention, convolution possesses an inductive bias that facilitates extracting high-frequency local information. However, convolutional feature aggregation solely relies on parameter-shared convolutional kernels, resulting in a lack of context-awareness. To address this issue, we utilize a self-modulating convolutional operator that embeds context-awareness into convolution, enabling it to extract local representation adaptively. Specifically, we generate context-aware weights through the $\mathrm{Sigmoid}$ and combine them with $\mathrm{DWConv}$ to adapively aggregate local information. This process is summarized in Eq. 8, where $\odot$ represents the Hadamard product, and $Q$ is directly derived from Eq. 7 for saving parameters of linear projection:

$$
\begin{aligned}
Q' &= \mathrm{DWConv}(Q), \\
X'_{local} &= Q' \odot \mathrm{Sigmoid}(Q') = Q' \odot \frac{1}{1 + e^{-Q'}}.
\end{aligned}
\tag{8}
$$

### 3.2.4 Bidirectional Adaptive Interaction

As illustrated in Fig. 1, when modeling the fox and its nose separately, two types of interactions - "Local to Global" and "Global to Local" - occur between their respective features. Inspired by the human visual system, we design a bidirectional adaptive interaction process that incorporates both types of interactions. To achieve bidirectional interaction, we adopt a method similar to the one described in Sec. 3.2.3 but utilize cross-modulation instead of self-modulation. Specifically, the equation for Local to Global interaction is given by Eq. 9:

$$
\begin{aligned}
X_{local} &= X'_{local} \odot \mathrm{Sigmoid}(X'_{global}) = X'_{local} \odot \frac{1}{1 + e^{-X'_{global}}} \\
&= Q' \odot \frac{1}{1 + e^{-Q'}} \odot \frac{1}{1 + e^{-X'_{global}}},
\end{aligned}
\tag{9}
$$

Similar to Eq. 9, Global to Local interaction is achieved by Eq. 10:

$$
\begin{aligned}
X_{global} &= X'_{global} \odot \mathrm{Sigmoid}(X'_{local}) = X'_{global} \odot \frac{1}{1 + e^{-X'_{local}}} \\
&= X'_{global} \odot \frac{1}{1 + e^{-Q' \odot \frac{1}{1+e^{-Q'}}}},
\end{aligned}
\tag{10}
$$

After completing the two interaction processes, the local and global representation contain information about each other. To merge these representation, we use point-wise multiplication and achieve intermingling among channels with a linear projection, as shown in Eq. 11:

$$
\begin{aligned}
Y &= W_2 \otimes (X_{global} \odot X_{local}) \\
&= W_2 \otimes (X'_{global} \odot Q' \odot \frac{1}{1 + e^{-Q'}} \odot \frac{1}{1 + e^{-X'_{global}}} \odot \frac{1}{1 + e^{-Q' \odot \frac{1}{1+e^{-Q'}}}})
\end{aligned}
\tag{11}
$$

It is worth noting that to get a faster and implementable model, we omit the last high-order variable in Eq. 11 and arrive at a concise expression. More details can be found in appendix. Through the combination of local adaptive aggregation, global adaptive aggregation, and bidirectional adaptive interaction, we present the comprehensive FASA module. As depicted in Fig. 3, our model, which draws inspiration from the human visual system, is both straightforward and simple. It adaptively models local, global, and the interaction between the two in context-aware manners. More discussion about bidirectional adaptive interaction can be found in the appendix.

## 4 Experiments

We conducted experiments on a wide range of vision tasks, including image classification on ImageNet-1K [9], object detection and instance segmentation on COCO 2017 [33], and semantic segmentation on ADE20K [80]. In addition to them, we also make ablation studies to validate the importance of each component. More details can be found in the appendix.

## 4.1 Image Classification

**Settings.** We train our models on ImageNet-1K [9] from scratch. And we follow the same training strategy in DeiT [54] for a fair comparison. Different from DeiT, we employ 20 epochs of linear warm-up, while in DeiT it is 5. The maximum rates of increasing stochastic depth [24] are set to 0.05/0.05/0.1/0.15 for B0/B1/B2/B3, respectively. More details can be found in the appendix.

| Size (M) | Model | Input | Params (M) | FLOPs (G) | Throughput (img/s) | Top-1 (%) | Size (M) | Model | Input | Params (M) | FLOPs (G) | Throughput (img/s) | Top-1 (%) |
|---|---|---|---|---|---|---|---|---|---|---|---|---|---|
| 0 ~ 5 | T2T-ViT-7 [73] | 224² | 4.3 | 1.1 | 1762 | 71.7 | 10 ~ 15 | XCiT-T24 [14] | 224² | 12.1 | 2.3 | 1194 | 79.4 |
| | QuadTree-B-b0 [52] | 224² | 3.5 | 0.7 | 885 | 72.0 | | ResT-S [78] | 224² | 13.7 | 1.9 | 918 | 79.6 |
| | TNT-Tiny [18] | 224² | 6.1 | 1.4 | 545 | 73.9 | | Shunted-T [47] | 224² | 11.5 | 2.1 | 957 | 79.8 |
| | Ortho-T [25] | 224² | 3.9 | 0.7 | — | 74.0 | | DeiT-S [54] | 224² | 22.1 | 4.6 | 899 | 79.9 |
| | EdgeViT-XXS [40] | 224² | 4.1 | 0.6 | 1926 | 74.4 | | QuadTree-B-b1 [52] | 224² | 13.6 | 2.3 | 543 | 80.0 |
| | MobileViT-XS [38] | 256² | 2.3 | 1.1 | 1367 | 74.8 | | RegionViT-Ti [2] | 224² | 13.8 | 2.4 | 710 | 80.4 |
| | LVT [66] | 224² | 5.5 | 0.9 | 1265 | 74.8 | | Wave-MLP-T [53] | 224² | 17.0 | 2.4 | 1052 | 80.6 |
| | CPVT-Ti-GAP [6] | 224² | 5.8 | 1.3 | — | 74.9 | | MPViT-XS [28] | 224² | 10.5 | 2.9 | 597 | 80.9 |
| | EdgeNeXt-XS [37] | 256² | 2.3 | 0.5 | 1417 | 75.0 | | EdgeViT-S [40] | 224² | 11.1 | 1.9 | 1049 | 81.0 |
| | PVT-T [57] | 224² | 13.2 | 1.6 | 1233 | 75.1 | | VAN-B1 [17] | 224² | 13.9 | 2.5 | 995 | 81.1 |
| | ViT-C [63] | 224² | 4.6 | 1.1 | — | 75.3 | | Swin-T [34] | 224² | 29.0 | 4.5 | 664 | 81.3 |
| | VAN-B0 [17] | 224² | 4.1 | 0.9 | 1662 | 75.4 | | CrossFormer-T [59] | 224² | 27.8 | 2.9 | 929 | 81.5 |
| | ViL-Tiny [77] | 224² | 6.7 | 1.4 | 857 | 76.3 | | ResT-B [78] | 224² | 30.3 | 4.3 | 588 | 81.6 |
| | CeiT-T [72] | 224² | 6.4 | 1.2 | — | 76.4 | | EfficientNet-B3 [51] | 300² | 12.0 | 1.8 | 634 | 81.6 |
| | FAT-B0 | 224² | 4.5 | 0.7 | 1932 | 77.6 | | FAT-B2 | 224² | 13.5 | 2.0 | 1064 | 81.9 |
| 5 ~ 10 | T2T-ViT-12 [73] | 224² | 6.9 | 1.9 | 1307 | 76.5 | 25 ~ 30 | DAT-T [61] | 224² | 29 | 4.6 | 577 | 82.0 |
| | Rest-lite [78] | 224² | 10.5 | 1.4 | 1123 | 77.2 | | FocalNet-T [67] | 224² | 29 | 4.4 | 610 | 82.1 |
| | XCiT-T12 [14] | 224² | 6.7 | 1.2 | 1676 | 77.1 | | Focal-T [68] | 224² | 29 | 4.9 | 301 | 82.2 |
| | EdgeViT-XS [40] | 224² | 6.7 | 1.1 | 1528 | 77.5 | | CrossFormer-S [59] | 224² | 31 | 4.9 | 601 | 82.5 |
| | CoaT-Lite-T [64] | 224² | 5.7 | 1.6 | 1045 | 77.5 | | RegionViT-S [2] | 224² | 31 | 5.3 | 460 | 82.6 |
| | SiT-Ti w/o FRD [83] | 224² | 15.9 | 1.0 | 1057 | 77.7 | | DaViT-T [10] | 224² | 28 | 4.5 | 616 | 82.8 |
| | RegNetY-1.6GF [44] | 224² | 11.2 | 1.6 | 1241 | 78.0 | | WaveViT-S [70] | 224² | 20 | 4.3 | 482 | 82.7 |
| | MPViT-T [28] | 224² | 5.8 | 1.6 | 737 | 78.2 | | QuadTree-B-b2 [52] | 224² | 24 | 4.5 | 299 | 82.7 |
| | MobileViT-S [38] | 256² | 5.6 | 2.0 | 898 | 78.4 | | CSWin-T [12] | 224² | 23 | 4.3 | 591 | 82.7 |
| | ParC-Net-S [76] | 256² | 5.0 | 1.7 | 1321 | 78.6 | | MPViT-S [28] | 224² | 23 | 4.7 | 410 | 83.0 |
| | PVTv2-B1 [58] | 224² | 13.1 | 2.1 | 1007 | 78.7 | | HorNet-T [46] | 224² | 23 | 4.0 | 586 | 83.0 |
| | PerViT-T [39] | 224² | 7.6 | 1.6 | 1402 | 78.8 | | FAT-B3-ST | 224² | 29 | 4.7 | 641 | 83.0 |
| | CoaT-Lite-Mi [64] | 224² | 11.0 | 2.0 | 963 | 79.1 | | ConvNeXt-S [35] | 224² | 50 | 8.7 | 405 | 83.1 |
| | EfficientNet-B1 [51] | 240² | 7.8 | 0.7 | 1395 | 79.2 | | LITv2-M [41] | 224² | 49 | 7.5 | 436 | 83.3 |
| | FAN-T-ViT [81] | 224² | 7.3 | 1.3 | 1181 | 79.2 | | EfficientFormer-L7 [30] | 224² | 82 | 10.2 | 368 | 83.3 |
| | EdgeNext-S [37] | 224² | 5.6 | 1.3 | 1243 | 79.4 | | Ortho-S [25] | 224² | 24 | 4.5 | – | 83.4 |
| | FAT-B1 | 224² | 7.8 | 1.2 | 1452 | 80.1 | | FAT-B3 | 224² | 29 | 4.4 | 474 | 83.6 |

Table 1: Comparison with the state-of-the-art on ImageNet-1K classification. "FAT-B3-ST" indicates the FAT-B3 with the same layout of Swin-T.

**Results.** We compare our FAT against many state-of-the-art models in Tab. 1. The comparison results demonstrate that our FAT consistently outperforms previous models across all settings, with a particularly significant improvement in performance on lightweight models. Specifically, our FAT-B0 achieves **77.6%** Top1-accuracy with only **4.5M** parameters and **0.7**GFLOPs. Our FAT-B1 also surpasses EdgeNext-S by **0.7%** with similar FLOPs. Furthermore, our FAT-B2 model surpasses CrossFormer-T in top1-accuracy by **0.4%** while using only half the parameters. Additionally, our FAT-B3 model has also outperformed many state-of-the-art models.

**Speed.** As shown in Tab. 1, we present the speed comparison results of all models on V100, and our FAT performs better than its counterparts in both speed and performance. Specifically, Our FAT-B0 has a 30% higher throughput than EdgeViT-XS, and a 0.1% higher top-1 accuracy compared to EdgeViT-XS.

| Backbone | Semantic FPN | | |
|---|---|---|---|
| | Params(M) | FLOPs(G) | mIoU(%) |
| PVTv2-B0 [58] | 7.6 | 25.0 | 37.2 |
| VAN-B0 [17] | 8.1 | 25.9 | 38.5 |
| EdgeViT-XXS [40] | 7.9 | 24.4 | 39.7 |
| FAT-B0 | 8.4 | 25.0 | 41.5 |
| PVT-T [57] | 17.0 | 33.2 | 35.7 |
| ResNet50 [20] | 28.5 | 45.6 | 36.7 |
| PoolFormer-S12 [71] | 16.2 | 31.0 | 37.2 |
| EdgeViT-XS [40] | 10.6 | 27.7 | 41.4 |
| FAT-B1 | 11.6 | 27.5 | 42.9 |
| ResNet18 [20] | 15.5 | 32.2 | 32.9 |
| PVTv2-B1 [58] | 17.8 | 34.2 | 42.5 |
| VAN-B1 [17] | 18.1 | 34.9 | 42.9 |
| FAT-B2 | 17.2 | 32.2 | 45.4 |
| Swin-T [34] | 31.9 | 182 | 41.5 |
| DAT-T [61] | 32.1 | 198 | 42.6 |
| ScalableViT [69] | 30.4 | 174 | 44.9 |
| CrossFormer-S [59] | 34.3 | 221 | 46.0 |
| VAN-B2 [17] | 30.3 | 164 | 46.7 |
| CSWin-T [12] | 26.1 | 202 | 48.2 |
| Ortho-S [25] | 28.0 | 195 | 48.2 |
| Shuted-S [47] | 26.1 | 183 | 48.2 |
| FAT-B3 | 32.9 | 179 | 48.9 |

Table 2: Comparison with the state-of-the-art on ADE20K.

## 4.2 Semantic Segmentation

**Settings.** We incorporate our pretrained FATs with the Semantic FPN [27]. As in [58; 57], we use $512 \times 512$ random crops during training and resize the images to have a shorter side of 512 pixels

during inference. Following [57], we fine-tune the models by AdamW with the learning rate of $1 \times 10^{-4}$ and batch size of 16. We train them for 80K iterations on the ADE20K training set.

**Results.** The results can be found in Tab. 2. For B0/B1/B2, the FLOPs are measured with input size of $512 \times 512$. While them for B3 are measured with $512 \times 2048$. All our models achieve the best performance in all comparisons. Specifically, our FAT-B1 exceeds EdgeViT-XS for **+1.5** mIoU. Moreover, our FAT-B3 outperforms the recent Shunted-S for **+0.7** mIoU. All the above results demonstrate our model's superiority in dense prediction.

## 4.3 Object Detection and Instance Segmentation

**Settings.** We conduct object detection and instance segmentation experiments on the COCO 2017 dataset [33]. Following Swin [34], we use the FAT family as the backbones and take RetinaNet [32] and Mask-RCNN [19] as detection and segmentation heads. We pretrain our backbones on ImageNet-1K and fine-tune them on the COCO training set with AdamW [36].

| Backbone | RetinaNet 1× | | | | | | | Mask R-CNN 1× | | | | | | |
|---|---|---|---|---|---|---|---|---|---|---|---|---|---|---|
| | Params(M) | $AP$ | $AP_{50}$ | $AP_{75}$ | $AP_S$ | $AP_M$ | $AP_L$ | Params(M) | $AP^b$ | $AP^b_{50}$ | $AP^b_{75}$ | $AP^m$ | $AP^m_{50}$ | $AP^m_{75}$ |
| DFvT-T [15] | - | - | - | - | - | - | - | 25 | 34.8 | 56.9 | 37.0 | 32.6 | 53.7 | 34.5 |
| PVTv2-B0 [58] | 13 | 37.2 | 57.2 | 39.5 | 23.1 | 40.4 | 49.7 | 24 | 38.2 | 60.5 | 40.7 | 36.2 | 57.8 | 38.6 |
| QuadTree-B-b0 [52] | 13 | 38.4 | 58.7 | 41.1 | 22.5 | 41.7 | 51.6 | 24 | 38.8 | 60.7 | 42.1 | 36.5 | 58.0 | 39.1 |
| EdgeViT-XXS [40] | 13 | 38.7 | 59.0 | 41.0 | 22.4 | 42.0 | 51.6 | 24 | 39.9 | 62.0 | 43.1 | 36.9 | 59.0 | 39.4 |
| FAT-B0 | 14 | 40.4 | 61.6 | 42.7 | 24.0 | 44.3 | 53.1 | 24 | 40.8 | 63.3 | 44.2 | 37.7 | 60.2 | 40.0 |
| DFvT-S [15] | - | - | - | - | - | - | - | 32 | 39.2 | 62.2 | 42.4 | 36.3 | 58.9 | 38.6 |
| EdgeViT-XS [40] | 16 | 40.6 | 61.3 | 43.3 | 25.2 | 43.9 | 54.6 | 27 | 41.4 | 63.7 | 45.0 | 38.3 | 60.9 | 41.3 |
| ViL-Tiny [77] | 17 | 40.8 | 61.3 | 43.6 | 26.7 | 44.9 | 53.6 | 27 | 41.4 | 63.5 | 45.0 | 38.1 | 60.3 | 40.8 |
| MPViT-T [28] | 17 | 41.8 | 62.7 | 44.6 | 27.2 | 45.1 | 54.2 | 28 | 42.2 | 64.2 | 45.8 | 39.0 | 61.4 | 41.8 |
| FAT-B1 | 17 | 42.5 | 64.0 | 45.1 | 26.9 | 46.0 | 56.7 | 28 | 43.3 | 65.6 | 47.4 | 39.6 | 61.9 | 42.8 |
| PVTv1-Tiny [57] | 23 | 36.7 | 56.9 | 38.9 | 22.6 | 38.8 | 50.0 | 33 | 36.7 | 59.2 | 39.3 | 35.1 | 56.7 | 37.3 |
| ResT-Small [78] | 23 | 40.3 | 61.3 | 42.7 | 25.7 | 43.7 | 51.2 | 33 | 39.6 | 62.9 | 42.3 | 37.2 | 59.8 | 39.7 |
| PVTv2-B1 [58] | 24 | 41.2 | 61.9 | 43.9 | 25.4 | 44.5 | 54.3 | 34 | 41.8 | 64.3 | 45.9 | 38.8 | 61.2 | 41.6 |
| DFvT-B [15] | - | - | - | - | - | - | - | 58 | 43.4 | 65.2 | 48.2 | 39.0 | 61.8 | 42.0 |
| QuadTree-B-b1 [52] | 24 | 42.6 | 63.6 | 45.3 | 26.8 | 46.1 | 57.2 | 34 | 43.5 | 65.6 | 47.6 | 40.1 | 62.6 | 43.3 |
| EdgeViT-S [40] | 23 | 43.4 | 64.9 | 46.5 | 26.9 | 47.5 | 58.1 | 33 | 44.8 | 67.4 | 48.9 | 41.0 | 64.2 | 43.8 |
| MPViT-XS [28] | 20 | 43.8 | 65.0 | 47.1 | 28.1 | 47.6 | 56.5 | 30 | 44.2 | 66.7 | 48.4 | 40.4 | 63.4 | 43.4 |
| FAT-B2 | 23 | 44.0 | 65.2 | 47.2 | 27.5 | 47.7 | 58.8 | 33 | 45.2 | 67.9 | 49.0 | 41.3 | 64.6 | 44.0 |
| Swin-T [34] | 38 | 41.5 | 62.1 | 44.2 | 25.1 | 44.9 | 55.5 | 48 | 42.2 | 64.6 | 46.2 | 39.1 | 61.6 | 42.0 |
| DAT-T [61] | 38 | 42.8 | 64.4 | 45.2 | 28.0 | 45.8 | 57.8 | 48 | 44.4 | 67.6 | 48.5 | 40.4 | 64.2 | 43.1 |
| DaViT-Tiny [10] | 39 | 44.0 | - | - | - | - | - | 48 | 45.0 | - | - | 41.1 | - | - |
| CMT-S [16] | 44 | 44.3 | 65.5 | 47.5 | 27.1 | 48.3 | 59.1 | 45 | 44.6 | 66.8 | 48.9 | 40.7 | 63.9 | 43.4 |
| MPViT-S [28] | 32 | 45.7 | 57.3 | 48.8 | 28.7 | 49.7 | 59.2 | 43 | 46.4 | 68.6 | 51.2 | 42.4 | 65.6 | 45.7 |
| QuadTree-B-b2 [52] | 35 | 46.2 | 67.2 | 49.5 | 29.0 | 50.1 | 61.8 | 45 | 46.7 | 68.5 | 51.2 | 42.4 | 65.7 | 45.7 |
| CSWin-T [12] | - | - | - | - | - | - | - | 42 | 46.7 | 68.6 | 51.3 | 42.2 | 65.6 | 45.4 |
| Shunted-S [47] | 32 | 45.4 | 65.9 | 49.2 | 28.7 | 49.3 | 60.0 | 42 | 47.1 | 68.8 | 52.1 | 42.5 | 65.8 | 45.7 |
| FAT-B3 | 39 | 45.9 | 66.9 | 49.5 | 29.3 | 50.1 | 60.9 | 49 | 47.6 | 69.7 | 52.3 | 43.1 | 66.4 | 46.2 |

Table 3: Comparison to other backbones using RetinaNet and Mask-RCNN on COCO val2017 object detection and instance segmentation.

**Results.** Tab. 3 shows the results with RetinaNet and Mask R-CNN. The results demonstrate that the proposed FAT performs best in all comparisons. For the RetinaNet framework, our FAT-B0 outperforms EdgeViT-XXS by **+0.7** AP, while B1/B2/B3 also perform better than their counterparts. As for the Mask R-CNN, our FAT-B3 outperforms the recent Shunted-S by **+0.5** box AP and **+0.6** mask AP. All the above results tell that our Fully Adaptive Transformer outperforms its counterparts by evident margins. More experiments can be found in appendix.

## 4.4 Comparison in Efficiency

We compared our approach with the state-of-the-art lightweight vision backbones, as shown in Tab 4. Our method achieved the best trade-off between speed and performance. The CPU is Intel i9Core and the GPU is V100. Inference throughput is measured with batch size 64. Inference latency is measured with batch size 1.

## 4.5 Larger Model

We scale up our FAT to 50M+ and 85M+ to match the general Vision Transformer backbones. For object detection/instance segmentation, all models use the framework of Mask-RCNN [19] with the 1x schedule. The results are shown in Tab. 5. It can be found that FAT has great scalability.

| Model | Params(M) | FLOPs(G) ↓ | CPU(ms) ↓ | GPU(ms) ↓ | Trp(imgs/s) ↑ | Top1-acc(%) |
|---|---|---|---|---|---|---|
| EdgeViT-XXS [40] | 4.1 | 0.6 | 43.0 | 14.2 | 1926 | 74.4 |
| MobileViT-XS [38] | 2.3 | 1.1 | 100.2 | 15.6 | 1367 | 74.8 |
| tiny-MOAT-0 [65] | 3.4 | 0.8 | 61.1 | 14.7 | 1908 | 75.5 |
| FAT-B0 | 4.5 | 0.7 | 44.3 | 14.4 | 1932 | 77.6 |
| EdgeViT-XS [40] | 6.7 | 1.1 | 62.7 | 15.7 | 1528 | 77.5 |
| ParC-Net-S [76] | 5.0 | 1.7 | 112.1 | 15.8 | 1321 | 78.6 |
| EdgeNext-S [37] | 5.6 | 1.3 | 86.4 | 14.2 | 1243 | 79.4 |
| FAT-B1 | 7.8 | 1.2 | 62.6 | 14.5 | 1452 | 80.1 |
| ParC-ResNet50 [76] | 23.7 | 4.0 | 160.0 | 16.6 | 1039 | 79.6 |
| tiny-MOAT-2 [65] | 9.8 | 2.3 | 122.1 | 15.4 | 1047 | 81.0 |
| EfficientNet-B3 [51] | 12.0 | 1.8 | 124.2 | 25.4 | 624 | 81.6 |
| FAT-B2 | 13.5 | 2.0 | 93.4 | 14.6 | 1064 | 81.9 |

Table 4: Comparison with other methods in efficiency.

| Model | Params(M) | FLOPs(G) | Top1-acc(%) | $AP^b$ | $AP^m$ |
|---|---|---|---|---|---|
| Swin-S [34] | 50 | 8.7 | 83.0 | 44.8 | 40.9 |
| Focal-S [68] | 51 | 9.1 | 83.5 | 47.4 | 42.8 |
| CMT-B [16] | 46 | 9.3 | 84.5 | – | – |
| FAT-B4 | 52 | 9.3 | 84.8 | 49.7 | 44.8 |
| CSwin-B [12] | 78 | 15.0 | 84.2 | – | – |
| MOAT-2 [65] | 73 | 17.2 | 84.7 | – | – |
| CMT-L [16] | 75 | 19.5 | 84.8 | – | – |
| FAT-B5 | 88 | 15.1 | 85.2 | – | – |

Table 5: Comparison with general backbones.

## 4.6 Ablation Study and Spectral Analysis

In this section, we conduct experiments to understand FAT better. The training settings are the same as in previous experiments. More experiments can be found in the appendix.

| | ImageNet-1K | | | COCO | | ADE20K |
|---|---|---|---|---|---|---|
| Model | Params(M) | FLOPs(G) | Top-1(%) | $AP^b$ | $AP^m$ | mIoU |
| add+linear | 4.5 | 0.72 | 76.2 | 39.0 | 35.8 | 39.6 |
| cat+linear | 4.8 | 0.77 | 76.6 | 39.6 | 36.3 | 40.2 |
| mul+linear | 4.5 | 0.72 | 77.1 | 40.3 | 37.1 | 40.9 |
| interaction | 4.5 | 0.72 | 77.6 | 40.8 | 37.7 | 41.5 |
| pool down | 4.4 | 0.71 | 77.2 | 40.2 | 36.9 | 40.6 |
| conv w/o overlap | 4.4 | 0.71 | 77.2 | 40.3 | 36.9 | 40.8 |
| conv w/ overlap | 4.5 | 0.71 | 77.3 | 40.5 | 37.3 | 40.9 |
| refined down | 4.5 | 0.72 | 77.6 | 40.8 | 37.7 | 41.5 |
| w/o conv. pos | 4.4 | 0.70 | 77.4 | 40.5 | 37.3 | 41.2 |
| conv. pos | 4.5 | 0.72 | 77.6 | 40.8 | 37.7 | 41.5 |

Table 6: Ablation of FAT

**Bidirectional Adaptive Interaction.** Initially, we validate the efficacy of bidirectional adaptive interaction by comparing it with three baselines for fusing local and global information: concatenation, addition and element-wise multiplication. As shown in Tab. 6, it can be observed that bidirectional adaptive interaction outperforms all baselines significantly across various tasks. Specifically, compared to the baseline, which uses the cat+linear to fuse the local and global information, our bidirectional adaptive interaction uses fewer parameters and FLOPs but surpasses the baseline by **1.0%** in Top1-accuracy. Our bidirectional adaptive interaction also performs better than element-wise multiplication.

**Fine-Grained Downsampling.** We compared our fine-grained downsampling strategy with three other strategies: non-overlapping large stride pooling, non-overlapping large stride convolution, and overlapping large stride convolution. As shown in Tab. 6, experimental results across various tasks have demonstrated the effectiveness of our fine-grained downsampling strategy. Specifically, our fine-grained downsampling strategy surpasses directly pooling downsampling strategy by **0.4%**

**Positional Encoding.** At the end of Tab. 6, we explore the effect of CPE, and the results in the table show that the flexible CPE also contributes to the performance improvement of the model for **0.2%** in image classification.

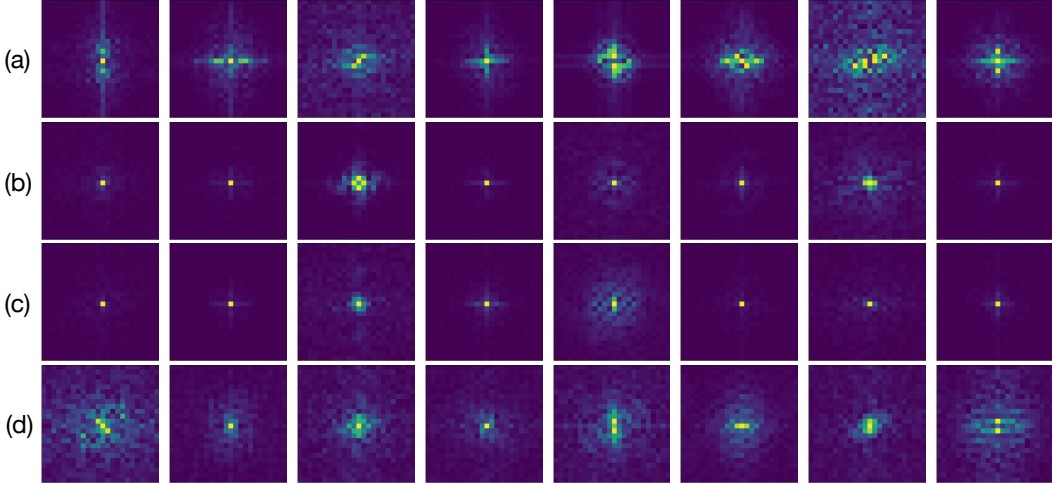

Figure 4: Spectral analysis from 8 output channels of FASA. The larger magnitude has a lighter color. Pixels that are closer to the center have a lower frequency. From top to bottom, the results are from (a) local adaptive aggregation, (b) global adaptive aggregation, (c) add+linear fusion, and (d) bidirectional adaptive interaction.

**Spectral Analysis.** As shown in Fig. 4, we conduct a spectral analysis on FASA. Compared to add+linear fusion, our bidirectional adaptive interaction can more fully fuse high-frequency and low-frequency information.

## 5 Conclusion

In this paper, we propose a new attention mechanism named Fully Adaptive Self-Attention (FASA), which draws inspiration from the human visual system. FASA is designed to model local and global information adaptively while also accounting for their bidirectional interaction using adaptive strategies. Furthermore, we enhance the self-attention mechanism in FASA by incorporating a fine-grained downsampling strategy to promote better global perception at a more detailed level. Using the flexible FASA as our foundation, we develop a lightweight vision backbone called Fully Adaptive Transformer (FAT), which can be applied across various vision tasks. Our extensive experiments on diverse asks, such as image classification, object detection, instance segmentation, and semantic segmentation, provide compelling evidence of the effectiveness and superiority of FAT. We expect to apply FAT to other vision-related pursuits, including video prediction and vision-language models.

## Acknowledgment

This work is partially funded by National Natural Science Foundation of China (Grant No. 62006228, U21B2045, U20A20223), Youth Innovation Promotion Association CAS (Grant No. 2022132), and Beijing Nova Program (20230484276).

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

# Appendix

## A    Discussion About "Interaction" And "Fusion"

Generally speaking, interaction refers to the process of two or more objects or systems affecting each other. In contrast, fusion refers to combining two or more things into a single entity. As shown in Fig. 5, in contrast to all previous feature fusion methods, our proposed bidirectional adaptive interaction stands out in two significant ways:

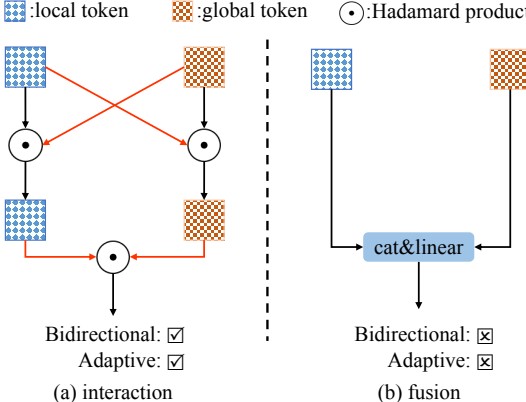

Figure 5: Comparison between our bidirectional adaptive interaction and traditional fusion method.

**1) Bidirectional Property:** Our approach draws inspiration from how visual information flows bidirectionally through the human visual system. Before fusing local and global features, we model a bidirectional process between the two. Specifically, for local features, we utilize global features to generate weights that encompass global information. These weights are applied to the local features, allowing them to have the ability to capture the global context. We execute a similar process for global features. This results in strengthened local and global features, respectively. In contrast to linear fusion, our interaction returns two enhanced vectors representing local and global contexts rather than an inseparable vector mixed with both [41; 48; 43].

**2) Adaptive Capability:** Taking local features as an example, the global weights we use to enhance them are not trainable. They are generated by tokens containing global features, making them context-aware and capable of adapting to the input data. Moreover, when fusing the enhanced local and global features, we do not introduce any additional parameters but calculate their element-wise product (Hadamard product) directly. Compared with linear fusion, our context-aware fusion approach is closer to the attention mechanism [56; 17; 22]. Our ablation experiments have shown that this fusion approach not only saves parameters but also achieves better results. Specifically, for a detailed comparison between the Add+Linear, Cat+Linear, and our model, please refer to the paper's main text.

The implementation of our proposed bidirectional adaptive interaction method is very simple, yet it has achieved remarkable results compared to other methods for fusing local and global features. Through this work, we aim to reveal the critical role that features fusion plays in visual models and hope to inspire more related research.

## B    More experiments

We evaluate our backbones with the framework of UperNet [62] based on MMsegmentation [7]. Following the [34], we adopt AdamW to optimize the model for 160K iterations. As shown in Tab. 7, our model surpasses all its counterparts and achieves the best performance. Specifically, our FAT-B3 surpass the resent Shunted-S by **+0.7** mIoU and **+0.8** MS mIoU.

| Backbone | Params(M) | FLOPs(G) | mIoU(%) | MS mIoU(%) |
|---|---|---|---|---|
| Swin-T [34] | 60 | 945 | 44.5 | 45.8 |
| DAT-T [61] | 60 | 957 | 45.5 | 49.0 |
| DaViT-T [10] | 60 | 940 | 46.3 | — |
| UniFormer-S [29] | 52 | 955 | 47.0 | 48.5 |
| CrossFormer-S [59] | 62 | 980 | 47.6 | 48.4 |
| MPViT-S [28] | 52 | 943 | 48.3 | — |
| Ortho-S [25] | 54 | 956 | 48.5 | 49.9 |
| Shunted-S [47] | 52 | 940 | 48.9 | 49.9 |
| FAT-B3 | 59 | 936 | 49.6 | 50.7 |

Table 7: Results with the framework of UperNet on ADE20K. The FLOPs are measured with the resolution of $512 \times 2048$.

| Output Size | Layer Name | FAT-B0 | FAT-B1 | FAT-B2 | FAT-B3 |
|---|---|---|---|---|---|
| $\frac{H}{4} \times \frac{W}{4}$ | Conv Stem | $3 \times 3, 16, s2$
$3 \times 3, 32, s2$
$3 \times 3, 32, s1$
$3 \times 3, 32, s1$
$1 \times 1, 32, s1$ | $3 \times 3, 24, s2$
$3 \times 3, 48, s2$
$3 \times 3, 48, s1$
$3 \times 3, 48, s1$
$1 \times 1, 48, s1$ | $3 \times 3, 32, s2$
$3 \times 3, 64, s2$
$3 \times 3, 64, s1$
$3 \times 3, 64, s1$
$1 \times 1, 64, s1$ | $3 \times 3, 32, s2$
$3 \times 3, 64, s2$
$3 \times 3, 64, s1$
$3 \times 3, 64, s1$
$1 \times 1, 64, s1$ |
| $\frac{H}{4} \times \frac{W}{4}$ | CPE
FASA
ConvFFN | $\begin{bmatrix} 3 \times 3, 32 \\ \text{kernel 3, heads 2} \\ 5 \times 5, 32 \end{bmatrix} \times 2$ | $\begin{bmatrix} 3 \times 3, 48 \\ \text{kernel 3, heads 3} \\ 5 \times 5, 48 \end{bmatrix} \times 2$ | $\begin{bmatrix} 3 \times 3, 64 \\ \text{kernel 3, heads 2} \\ 5 \times 5, 64 \end{bmatrix} \times 2$ | $\begin{bmatrix} 3 \times 3, 64 \\ \text{kernel 3, heads 2} \\ 5 \times 5, 64 \end{bmatrix} \times 4$ |
| $\frac{H}{8} \times \frac{W}{8}$ | CPE
FASA
ConvFFN | $\begin{bmatrix} 5 \times 5, 80 \\ \text{kernel 5, heads 5} \\ 5 \times 5, 80 \end{bmatrix} \times 2$ | $\begin{bmatrix} 5 \times 5, 96 \\ \text{kernel 5, heads 6} \\ 5 \times 5, 96 \end{bmatrix} \times 2$ | $\begin{bmatrix} 5 \times 5, 128 \\ \text{kernel 5, heads 4} \\ 5 \times 5, 128 \end{bmatrix} \times 2$ | $\begin{bmatrix} 5 \times 5, 128 \\ \text{kernel 5, heads 4} \\ 5 \times 5, 128 \end{bmatrix} \times 4$ |
| $\frac{H}{16} \times \frac{W}{16}$ | CPE
FASA
ConvFFN | $\begin{bmatrix} 7 \times 7, 160 \\ \text{kernel 7, heads 10} \\ 5 \times 5, 160 \end{bmatrix} \times 6$ | $\begin{bmatrix} 7 \times 7, 192 \\ \text{kernel 7, heads 12} \\ 5 \times 5, 192 \end{bmatrix} \times 6$ | $\begin{bmatrix} 7 \times 7, 256 \\ \text{kernel 7, heads 8} \\ 5 \times 5, 256 \end{bmatrix} \times 6$ | $\begin{bmatrix} 7 \times 7, 256 \\ \text{kernel 7, heads 8} \\ 5 \times 5, 256 \end{bmatrix} \times 16$ |
| $\frac{H}{32} \times \frac{W}{32}$ | CPE
FASA
ConvFFN | $\begin{bmatrix} 9 \times 9, 256 \\ \text{kernel 9, heads 16} \\ 5 \times 5, 256 \end{bmatrix} \times 2$ | $\begin{bmatrix} 9 \times 9, 384 \\ \text{kernel 9, heads 24} \\ 5 \times 5, 384 \end{bmatrix} \times 2$ | $\begin{bmatrix} 9 \times 9, 512 \\ \text{kernel 9, heads 16} \\ 5 \times 5, 512 \end{bmatrix} \times 2$ | $\begin{bmatrix} 9 \times 9, 512 \\ \text{kernel 9, heads 16} \\ 5 \times 5, 512 \end{bmatrix} \times 4$ |
| $1 \times 1$ | Classifier | Fully Connected Layer, 1000 | | | |
| | Params | 4.5M | 7.8M | 13.5M | 29M |
| | FLOPs | 0.7G | 1.2G | 2.0G | 4.4G |

Table 8: Details about FAT's architecture.

## C   Implementation Details

### C.1   Architecture Details

The detailed architectures are shown in Tab. 8, where all FLOPs are measured at the resolution of $224 \times 224$ for image classification. For the convolution stem, we adopt four $3 \times 3$ convolutions and one $1 \times 1$ convolution to tokenize the image. Batch normalization and ReLU are used after each $3 \times 3$ convolution, and a layer normalization is used after the final $1 \times 1$ convolution. Inspired by the fact that in the vision backbone, the early stages tend to capture high-frequency local information, and the later stages tend to capture low-frequency global information, we gradually increase the DWConv's kernel sizes used in FASA. Specifically, we set the kernel size to 3 in the first stage and 9 in the last stage. The expansion ratios are set to 4, and the kernel sizes are set to 5 for all ConvFFN layers.

### C.2   Image Classification

We follow the same training strategy with Swin-Transformer [34]. All models are trained for 300 epochs from scratch. To train the models, we use the AdamW optimizer with a cosine decay learning rate scheduler and 20 epoch linear warm-up. We set the initial learning rate, weight decay, and batch size to 0.001, 0.05, and 1024, respectively. We adopt the strong data augmentation and regularization used in [34]. Our settings are RandAugment [8] (randm9-mstd0.5-inc1), Mixup [75] (prob=0.8), Cut-Mix [74] (prob=1.0), Random Erasing [79] (prob=0.25), increasing stochastic depth [24] (prob=0.05, 0.1, 0.1, 0.15 for FAT-B0, FAT-B1, FAT-B2 and FAT-B3).

## C.3 Object Detection and Instance Segmentation

We adopt MMDetection [3] to implement RetinaNet [32] and Mask-RCNN [19]. We use the commonly used "1×" (12 training epochs) setting for the two strategies. Following [34], during training, images are resized to the shorter side of 800 pixels while the longer side is within 1333 pixels. We adopt the AdamW optimizer with a learning rate of 0.0001, weight decay of 0.05, and batch size of 16 to optimize the model. The learning rate declines with the decay rate of 0.1 at the epoch 8 and 11.

## C.4 Semantic Segmentation

We adopt the Semantic FPN [27] and UperNet [62] based on MMSegmentation [7] and apply FAT pretrained on ImageNet-1K as backbone. We use the same setting of PVT [57] to train the Semantic FPN, and we use the "1×" schedule (trained for 80k iterations). All the models are trained with the input resolution of $512 \times 512$. When testing the model, we resize the shorter side of the image to 512 pixels. As for UperNet, we follow the default settings in Focal Transformer [68]. We take AdamW with a weight decay of 0.01 as the optimizer and train the models for 160K iterations. The learning rate is set to $6 \times 10^{-5}$ with 1500 iterations warmup.

# D More Ablation and Analysis Results

| Model | Blocks | Channels | Params(M) | FLOPs(G) | Top-1 acc(%) |
|---|---|---|---|---|---|
| Swin-T [34] | [2, 2, 6, 2] | [96, 192, 384, 768] | 29 | 4.5 | 81.3 |
| DAT-T [61] | [2, 2, 6, 2] | [96, 192, 384, 768] | 29 | 4.6 | 82.0 |
| FocalNet-T [67] | [2, 2, 6, 2] | [96, 192, 384, 768] | 28 | 4.4 | 82.1 |
| Focal-T [68] | [2, 2, 6, 2] | [96, 192, 384, 768] | 29 | 4.9 | 82.2 |
| FAT-B3-ST | [2, 2, 6, 2] | [96, 192, 384, 768] | 29 | 4.7 | **83.0** |

Table 9: Comparison with four baseline models when use the same layout with Swin-T.

**Other architecture design choices.** We modified FAT-B3 to have the same layout as Swin-T and compared it to four baseline models: Swin-T [34], DAT-T [61], FocalNet-T [67] and Focal-T [68]. As shown in Tab. 9, our model demonstrates significant superiority, with an accuracy 1.7% higher than Swin-T and 0.8% higher than Focal-Tiny.

**Contribution Isolation.** To better isolate the contribution of our FASA, we compare the different self-attention mechanisms on the same backbone. We choose five kinds of self-attention mechanisms to make the comparison: CSWSA in CSwin-Transformer [12], Max-SA in MaxViT [55], WSA/S-WSA in Swin-Transformer [34], SRA in PVT [57] and LSA/GSA in Twins-SVT [5]. The results are shown in Tab. 10. The networks' layout and the setting of CPE and ConvFFN are the same as FAT-B0. It can be found that our FASA surpasses all its counterparts. Compared to the latest Max-SA, our FASA achieved a gain of 1.9%.

| Method | Params (M) | FLOPs (G) | Top1-acc (%) |
|---|---|---|---|
| Max-SA [55] | 4.3 | 0.8 | 75.7 |
| WSA/S-WSA [34] | 4.3 | 0.8 | 76.1 |
| SRA [57] | 4.4 | 0.7 | 76.2 |
| LSA/GSA [5] | 4.3 | 0.7 | 76.6 |
| CSWSA [12] | 4.3 | 0.8 | 76.8 |
| FASA | 4.5 | 0.7 | **77.6** |

Table 10: Comparison among different self-attention mechanisms.

**Local Aggregation in FASA.** To validate that the model's capability to perceive local context does not solely rely on CPE and ConvFFN, we conducted experiments by removing local adaptive aggregation and bidirectional adaptive interaction from FASA, retaining only global adaptive aggregation. The results presented in Tab. 11 demonstrate a significant decrease in model performance (**1.4%**) when using only global adaptive aggregation. This outcome convincingly supports the soundness of our proposed FASA model.

| local aggregation | Params (M) | FLOPs (G) | Top1-acc (%) |
|---|---|---|---|
| w/o local | 4.4 | 0.70 | 76.2 |
| local | 4.5 | 0.72 | **77.6** |

Table 11: Ablation of kernel size used in FASA.

**Additional Sigmoid.** In our main text, in the Method section's Bidirectional Adaptive Interaction subsection, Eq. 11 omits the high-order variable:

$$\frac{1}{1 + e^{-Q' \odot \frac{1}{1+e^{-Q'}}}}$$

| Additional Sigmoid | Params (M) | FLOPs (G) | Throughput (imgs/s) | Top1-acc (%) |
|---|---|---|---|---|
| Sigmoid | 4.5 | 0.72 | 1815 | 77.6 |
| w/o Sigmoid | 4.5 | 0.72 | 1932 | 77.6 |

Table 12: Ablation of addtional Sigmoid.

in the final expression to a faster implementation. This variable represents two concatenated Sigmoid activation functions within the local adaptive aggregation phase. We attempt to include the previously omitted activation function in this section but find that having too many activation functions results in a similar performance but slower speed, as shown in Tab. 12. This result confirms the rationality of the structure we designed.

**Kernel Size of FASA.** Inspired by the notion that early stages of vision backbones tend to aggregate lower-level local features, while later stages prefer higher-level semantic ones, we adopt a staged approach and incrementally increase the DWConv's kernel size in FASA to aggregate local features. In this section, we investigate how DW-Conv's kernel size in FASA affects the model's image classification performance by varying the kernel sizes in different stages, as depicted in Tab. 13. Our experimental results demonstrate the effectiveness of our strategy for gradually increasing kernel size at different stages. Models using kernel sizes of 3, 5, 7, 9, and 3, 5, 7, 7 achieved the best performance.

| kernel sizes | Params (M) | FLOPs (G) | Top1-acc (%) |
|---|---|---|---|
| 3, 3, 3, 3 | 4.3 | 0.697 | 77.3 |
| 3, 5, 7, 7 | 4.5 | 0.717 | **77.6** |
| 3, 5, 7, 9 | 4.5 | 0.718 | **77.6** |
| 7, 7, 7, 7 | 4.5 | 0.737 | 77.5 |
| 9, 9, 9, 9 | 4.6 | 0.770 | 77.4 |

Table 13: Ablation of kernel size used in FASA.

**Convolution Stem.** We use the convolution stem proposed in [63]. To investigate the impact of convolution stem on our model's performance, we conducted experiments and presented the results in Tab. 14. Despite incurring some extra computational overhead, using convolution stem can significantly enhance the top1 accuracy of the model by **0.4%**.

| conv stem | Params (M) | FLOPs (G) | Top1-acc (%) |
|---|---|---|---|
| w/o stem | 4.5 | 0.65 | 77.2 |
| stem | 4.5 | 0.72 | **77.6** |

Table 14: Ablation of convolution stem.

**Kernel Size of ConvFFN.** We conduct ablation experiments to validate the effect of different kernel sizes for DWConv in ConvFFN. As shown in Tab. 15, the kernel size of $7 \times 7$ achieves the best performance, but it consumes more parameters (+0.2M) and FLOPs (+0.03G). To achieve a better trade-off between the performance and computational complexity, We set the kernel size of DWConv in ConvFFN to 5.

| kernel sizes | Params (M) | FLOPs (G) | Top1-acc (%) |
|---|---|---|---|
| $3 \times 3$ | 4.4 | 0.69 | 77.4 |
| $5 \times 5$ | 4.5 | 0.72 | **77.6** |
| $7 \times 7$ | 4.7 | 0.75 | 77.7 |

Table 15: Ablation of kernel size used in ConvFFN.

**Bidirectional Interaction.** Here we ablate the local-to-global and global-to-local, respectively, to show the effect of bidirectional interaction. The results are shown in the Tab. 16. It can be seen that the bidirectional interaction plays an important role in FAT.

# E  Speed Measurement

In order to accelerate the inference speed of our model, we utilized a toolkit developed for accelerating large kernel DW-Conv in [11]. Additionally, we employed the technique of structural reparameterization to merge the residual blocks in ConvFFN into one convolutional block. Furthermore, after omitting

| Model | Params (M) | FLOPs (G) | Top1-acc (%) |
|---|---|---|---|
| no interaction | 4.5 | 0.72 | 76.2 |
| w/o global to local | 4.5 | 0.72 | 76.9 |
| w/o local to global | 4.5 | 0.72 | 76.8 |
| bidirectional interaction | 4.5 | 0.72 | 77.6 |

Table 16: Ablation of bidirectional interaction.

the higher-order variables in bidirectional adaptive interaction, the original expression can be simplified to two SiLU activations followed by the multiplication of two branches. These measures significantly accelerated the inference speed of our model. Here, we list the most recent models in Tab. 17. As can be seen from the table, our model performs the best in terms of both speed and performance.

| Model | Throughput (imgs/s) | Top1-Acc (%) | Year | Model | Throughput (imgs/s) | Top1-Acc (%) | Year |
|---|---|---|---|---|---|---|---|
| T2T-ViT-7 [73] | 1762 | 71.7 | ICCV'2021 | Swin-T [34] | 664 | 81.3 | ICCV'2021 |
| QuadTree-B-b0 [52] | 885 | 72.0 | ICLR'2022 | PoolFormer-S36 [71] | 601 | 81.4 | CVPR'2022 |
| TNT-Tiny [18] | 545 | 73.9 | NeurIPS'2021 | CrossFormer-T [59] | 929 | 81.5 | ICLR'2022 |
| EdgeViT-XXS [40] | 1926 | 74.4 | ECCV'2022 | EfficientNet-B3 [51] | 634 | 81.6 | ICML'2019 |
| LVT [66] | 1265 | 74.8 | CVPR'2022 | ResT-Base [78] | 588 | 81.6 | NeurIPS'2021 |
| MobileViT-XS [38] | 1367 | 74.8 | ICLR'2022 | **FAT-B2** | **1064** | **81.9** | **Ours** |
| EdgeNeXt-XS [37] | 1417 | 75.0 | ECCVW'2022 | Coat-Lite-S [64] | 540 | 81.9 | ICCV'2021 |
| VAN-B0 [17] | 1662 | 75.4 | Arxiv'2022 | DAT-T [61] | 577 | 82.0 | CVPR'2022 |
| ViL-Tiny [77] | 857 | 76.7 | ICCV'2021 | PVTv2-B2 [58] | 582 | 82.0 | Arxiv'2022 |
| PoolFormer-S12 [71] | 1722 | 77.2 | CVPR'2022 | PiT-B [21] | 328 | 82.0 | ICCV'2021 |
| XCiT-T12 [14] | 1676 | 77.1 | NeurIPS'2021 | FocalNet-T [67] | 610 | 82.1 | NeurIPS'2022 |
| ResT-Lite [78] | 1123 | 77.2 | NeurIPS'2021 | ViL-S [77] | 359 | 82.4 | ICCV'2021 |
| **FAT-B0** | **1932** | **77.6** | **Ours** | CrossFormer-S [59] | 601 | 82.5 | ICLR'2022 |
| T2T-ViT-12 [73] | 1307 | 76.5 | ICCV'2021 | RegionViT-S [2] | 460 | 82.6 | ICLR'2022 |
| EdgeViT-XS [40] | 1528 | 77.5 | ECCV'2022 | WaveMLP-S [53] | 599 | 82.6 | CVPR'2022 |
| CoaT-Lite-T [64] | 1045 | 77.5 | ICCV'2021 | CSWin-T [12] | 591 | 82.7 | CVPR'2022 |
| SiT w/o FRD [83] | 1057 | 77.7 | ECCV'2022 | WaveViT-S [70] | 482 | 82.7 | ECCV'2022 |
| RegNetY-1.6GF [44] | 1241 | 78.0 | CVPR'2020 | VAN-B2 [17] | 531 | 82.8 | Arxiv'2022 |
| MPViT-T [28] | 737 | 78.2 | CVPR'2022 | DaViT-T [10] | 616 | 82.8 | ECCV'2022 |
| MobileViT-S [38] | 898 | 78.4 | ICLR'2022 | HorNet-T [46] | 586 | 82.8 | NeurIPS'2022 |
| ParC-Net-S [76] | 1321 | 78.6 | ECCV'2022 | FAN-S-ViT [81] | 525 | 82.9 | ICML'2022 |
| PVTv2-B1 [58] | 1007 | 78.7 | Arxiv'2022 | **FAT-B3-ST** | **641** | **83.0** | **Ours** |
| PerViT [39] | 1402 | 78.8 | NeurIPS'2022 | DeiT-B [54] | 299 | 81.8 | ICML'2021 |
| CoaT-Lite-Mi [64] | 963 | 79.1 | ICCV'2021 | Focal-T [68] | 301 | 82.2 | NeurIPS'2021 |
| EfficientNet-B1 [51] | 1395 | 79.2 | ICML'2019 | PoolFormer-M48 [71] | 304 | 82.5 | CVPR'2022 |
| FAN-T-ViT [81] | 1181 | 79.2 | ICML'2022 | QuadTree-B-b2 [52] | 299 | 82.7 | ICLR'2022 |
| EdgeNeXt-S [37] | 1243 | 79.4 | ECCVW'2022 | Shunted-S [47] | 456 | 82.9 | CVPR'2022 |
| XCiT-T24 [14] | 1194 | 79.4 | NeurIPS'2021 | MPViT-S [28] | 410 | 83.0 | CVPR'2022 |
| **FAT-B1** | **1452** | **80.1** | **Ours** | Swin-S [34] | 390 | 83.0 | ICCV'2021 |
| ResT-S [78] | 918 | 79.6 | NeurIPS'2021 | ConvNeXt-S [35] | 405 | 83.1 | CVPR'2022 |
| Shunted-T [47] | 957 | 79.8 | CVPR'2022 | PVTv2-B3 [58] | 398 | 83.2 | Arxiv'2022 |
| DeiT-S [54] | 899 | 79.9 | ICML'2021 | RegionViT-B [2] | 256 | 83.2 | ICLR'2022 |
| QuadTree-B-b1 [52] | 543 | 80.0 | ICLR'2022 | EfficientFormer-L7 [30] | 368 | 83.3 | NeurIPS'2022 |
| RegionViT-Ti [2] | 710 | 80.4 | ICLR'2022 | LITv2-M [41] | 436 | 83.3 | NeurIPS'2022 |
| WaveMLP-T [53] | 1052 | 80.6 | CVPR2022 | FocalNet-S [67] | 358 | 83.4 | NeurIPS'2022 |
| PiT-S [21] | 1042 | 80.9 | ICCV'2021 | iFormer-S [48] | 471 | 83.4 | NeurIPS'2022 |
| MPViT-XS [28] | 597 | 80.9 | CVPR'2022 | CrossFormer-B [59] | 368 | 83.4 | ICLR'2022 |
| EdgeViT-S [40] | 1049 | 81.0 | ECCV'2022 | WaveMLP-M [53] | 359 | 83.4 | CVPR'2022 |
| VAN-B1 [17] | 995 | 81.1 | Arxiv'2022 | **FAT-B3** | **474** | **83.6** | **Ours** |

Table 17: Comparisons of speed and performance among different models. All speeds are measured on the V100 32G with the batch size of 64.

## F  Limitations and Broader Impacts

While our FAT achieves superior performance with fast processing speed, one limitation of our work is that due to computational resource constraints, we do not apply our lightweight vision backbone to unsupervised learning, video processing, or visual language tasks. Additionally, we do not pretrain our models on large-scale datasets like ImageNet-21K. However, we look forward to exploring more applications of our proposed FAT in the future.

This study is purely academic in nature, and we are unaware of any negative social impact resulting directly from our work. However, we acknowledge that our models' potential malicious use is a concern affecting the entire field. Discussions related to this matter are beyond the scope of our research.

