# OpenReview forum: "Lightweight Vision Transformer with Bidirectional Interaction"
_NeurIPS.cc/2023/Conference — NeurIPS 2023 poster_

### Official Review · Reviewer_DXVA · 2023-06-29

**Soundness:** 3 good
**Presentation:** 3 good
**Contribution:** 2 fair
**Rating:** 6
**Confidence:** 5

**Summary:**

The paper proposes a new lightweight ViT structure called FAT. They use a fully adaptive self-attention mechanism for vision transformer to model the local and global information as well as the bidirectional interaction between them in context-aware ways. In addition, the paper introduces a fine-grained downsampling strategy to enhance the down-sampled self-attention mechanism for finer-grained global perception capability.

**Strengths:**

- The paper is well-written and easy to understand.

- The experimental results show the effectiveness of the method.

**Weaknesses:**

- The usage of conv stem, conditional positional encoding, the convFFN and the aggregation of global and local attention is not novel. Thus, the novelty of this paper primarily lies in using bidirectional interaction. However, it is un-natural to explain via human visual system of using local to global and global to local interaction. Is it really the way human look at pictures?

- In Eq.5, it is unclear why using different A1 and A2 makes the interaction 'bidirectional'. The author should further explain this.

- Lack of comparison to the state-of-the-art models. For example, the proposed FAT-B3 has the same accuracy to CMT-S [1]  but larger FLOPs and parameters.

[1] CMT: Convolutional Neural Networks Meet Vision Transformers. CVPR 2022.

**Questions:**

See weaknesses above.

**Limitations:**

See weaknesses above.

---

> ### Author Rebuttal · Authors · 2023-08-08
>
> ### **On motivation & novelty, explanation of Eq.5 and Comparison With SOTA**
>
> **Q1:** Motivation & Novelty
>
> **R1:** Thanks. The key novelty of our paper is bidirectional interaction, which is simple to implement and has negligible costs. This module is inspired by existing research on the human brain. In [1], it has been indicated that the human brain has the ability to interpret local features as global shapes. Additionally, in [2], it has been pointed out that the global coordination of local information significantly affects the processing of the human mid-level visual cortex. These two behaviors align with the examples mentioned in our Introduction, suggesting that the information flows of local to global and global to local should objectively exist in the human brain. Based on these perspectives, we separately model these two types of information flows in our bidirectional interaction. Since we aim to design an efficient visual backbone, we did not employ more complex modeling techniques. It has been proven that even with this simple bi-directional interaction pattern, the model can provide significant improvements. In contrast to all previous feature fusion methods, our bidirectional adaptive interaction stands out in two significant ways:
>
> 1. Bidirectional Property: We model a bidirectional process between the two before fusing local and global features. This results in strengthened local and global features, respectively. In contrast to commonly used linear fusion, our interaction returns two enhanced vectors representing local and global contexts rather than an inseparable vector mixed with both.
> 2. Adaptive Capability: Taking local features as an example, the global weights we use to enhance them are generated by tokens containing global features, making them context-aware and capable of adapting to the input data. Moreover, we calculate their element-wise product directly when fusing the enhanced local and global features. Our context-aware fusion approach is closer to the attention mechanism than linear fusion.
>
>
>
> **Q2:** Explanation of Eq.5.
>
> **R2:** Thanks, Eq.5 can be written as $y_i=\mathcal{M}(\mathcal{F}(\mathcal{A}_1(x_i, X)), \mathcal{A}_2(x_i, X))$, $\mathcal{A}_1$ and $\mathcal{A}_2$ denote two different feature aggregation operators, $\mathcal{F}$ is activation function, and $\mathcal{M}$ is modulation operator. Our bidirectional interaction includes two processes. Consider the local to global process. The global features and local features have been aggregated by attention ($\mathcal{A}_1$) and self-modulated convolution ($\mathcal{A}_2$), respectively. Then we use ${\rm Sigmoid}$ ($\mathcal{F}$) to activate global features to generate global weights. After that, we use global weights to modulate local features. The whole local to global peocess can be represented by $y_i=\mathcal{M}(\mathcal{F}(\mathcal{A}_1(x_i, X)), \mathcal{A}_2(x_i, X))$. Similarly, The global to local process also can be written as this equation, except $\mathcal{A}_1$ and $\mathcal{A}_2$ swap their positions. For simplicity, we only use one equation to denote these two processes in our paper. Here, we present the complete formulation of these two processes. Thus the complete bidirectional interaction can be expressed as:
>
> $y_{i2}=\mathcal{M}(\mathcal{F}(\mathcal{A}_1(x_i, X)), \mathcal{A}_2(x_i, X))$
>
>
> $y_{i1}=\mathcal{M}(\mathcal{F}(\mathcal{A}_2(x_i, X)), \mathcal{A}_1(x_i, X))$
>
> These two processes together constitute bidirectional interaction. We thank the reviewer for pointing out this point, and we will modify it in the next version of the paper.
>
> **Q3:** Comparison with SOTA.
>
> **R3:** Thanks. Here we scale up FAT to the general backbone and compare it with recent SOTA. We also use the framework of Mask-RCNN to conduct object detection/instance segmentation with the 1x schedule. It can be seen that FAT performs better than CMT, especially in downstream tasks (object detection and instance segmentation). And when it comes to the larger model, the performance advantage of FAT will be further enhanced.
>
> |Model|Params(M)|FLOPs(G)|Top1-acc(%)|$AP^b$|$AP^m$|
> |--|--|--|--|--|--|
> |Swin-T|29|4.5|81.3|42.2|39.1|
> |UniFormer-S|24|4.2|82.9|45.6|41.6|
> |MOAT-0 [3]|28|5.7|83.3|--|--|
> |CMT-S [4]|25|4.0|83.5|44.6|40.7|
> |FAT-B3|29|4.4|**83.6**|**47.6**|**43.1**|
> |||||||
> |||||||
> |Swin-S|50|8.7|83.0|44.8|40.9|
> |Focal-S|51|9.1|83.5|47.4|42.8|
> |UniFormer-B|50|8.3|83.9|47.4|43.1|
> |MOAT-1 [3]|42|9.1|84.2|--|--|
> |CMT-B [4]|46|9.3|84.5|--|--|
> |FAT-B4|52|9.3|**84.8**|**49.7**|**44.8**|
> |||||||
> |||||||
> |CSwin-B|78 |15.0|84.2|--|--|
> |MOAT-2 [3]|73|17.2|84.7|--|--|
> |CMT-L [4]|75|19.5|84.8|--|--|
> |iFormer-L |87|14.0|84.8|--|--|
> |FAT-B5|88|15.1|**85.2**|--|--|
>
> ****
>
> **Reference**
>
> [1]Schwarzkopf D. Samuel and Rees Geraint. 2011 Interpreting local visual features as a global shape requires awareness Proc. R. Soc. B.278: 2207–2215. doi/10.1098/rspb.2010.1909
>
> [2]Mannion DJ, Kersten DJ, Olman CA. Regions of mid-level human visual cortex sensitive to the global coherence of local image patches. J Cogn Neurosci. 2014 Aug;26(8):1764-74. doi: 10.1162/jocn_a_00588. Epub 2014 Feb 24. PMID: 24564470; PMCID: PMC4074231.
>
> [3]Chenglin Yang, et al. Moat: alternating mobile convolution and attention brings strong vision models. ICLR, 2023.
>
> [4]Jianyuan Guo, et al. Cmt: Convolutional neural networks meet vision transformers. CVPR, 2022.

---

> > ### Comment · Reviewer_DXVA · 2023-08-14
> > **Post review**
> >
> > I appreciate the authors for dealing with my concerns and including more experimental results. All of my questions are well resolved and this makes me more inclined to increase my score.

---

> > > ### Author Response · Authors · 2023-08-15
> > > **Thanks for the reviewer DXVA's comments**
> > >
> > > Dear reviewer DXVA,
> > >
> > > Thank you very much for your recognition and appreciation of our work. We sincerely appreciate the time and effort you have put into reviewing our paper. Your feedback and support are crucial in improving our work. Furthermore, we highly value your thorough reading and detailed comments.
> > >
> > > Best regards,
> > >
> > > The authors

---

### Official Review · Reviewer_cvP4 · 2023-07-06

**Soundness:** 3 good
**Presentation:** 3 good
**Contribution:** 3 good
**Rating:** 6
**Confidence:** 3

**Summary:**

This paper presents a new family of lightweight vision transformers that are built on the Fully Adaptive Self-Attention module. The key innovation of these transformers lies in their bidirectional interactive module, which enhances both local and global features. The experiments primarily focus on image classification, object detection, and semantic segmentation tasks. Extensive evaluations were conducted on various common benchmarks, including ImageNet-1K, to assess the performance of the proposed transformers.

**Strengths:**

The proposed method demonstrates a key innovation through bidirectional interaction between the global and local branches, leading to significant improvement as validated by comprehensive ablation studies. The study includes extensive experiments, and the ablation analysis is thorough, highlighting the quality of the research. The writing is clear and concise, effectively presenting the experimental setting and demonstration. Impressively, FAT-B0 and B1 achieve remarkable results in terms of both throughput and accuracy, providing valuable insights for future research on low-latency small models.

**Weaknesses:**

1. The scaling behavior of the method is my primary concern. While I appreciate the author's focus, the current results indicate that larger models exhibit limited improvement over the baseline in the three tasks. In my opinion, it is essential to discuss the scaling aspect for a proposed general backbone.
2. The second issue is the lack of comparison in terms of inference time between different methods or different hardware (e.g. CPU/GPU). Based on the current results, the proposed model works very well under low parameter settings, providing the direct benefit of lighter and more efficient deployment on mobile devices. However, the author has not provided sufficient evidence to demonstrate whether the proposed method maintains efficient inference efficiency with such low computational requirements. This aspect is crucial for me in proposing a lightweight model because training speed is often not a concern for these types of models. Instead, focusing on comparing the capabilities of this model under different hardware and computational costs can significantly enhance the significance of this method.
3. The motivation for this paper appears somewhat forced, particularly from Figure 1. I have not come across any related work discussing this phenomenon. If any such works exist, please cite them, as it would enhance the trustworthiness of your motivation.


**Questions:**

1. My first two questions are motivated by different considerations, and I recommend that the author concentrate on one aspect to augment the method's significance. Given the exceptional performance of the proposed model on small-scale counterparts, my particular interest lies in contrasting the inference across different hardware and CNN architectures. Nevertheless, should the author contend that the proposed model should possess broader applicability, I am also intrigued to observe its scaling behavior.
2. I am interested in investigating the training stability of the proposed methods. It would be helpful if there are any plots illustrating the training stability.


**Limitations:**

The author has discussed their limitation in the paper.

---

> ### Author Rebuttal · Authors · 2023-08-08
>
> ### **On scale-up capability, efficiency, motivation, and training stability**
>
> We thank the reviewer for recognizing the positive aspects of our paper, and we will address the reviewer's concerns in the following parts.
>
> **Q1:** scaling behavior of the method.
>
> **R1:** Thanks, we scale up our FAT to 25M+, 50M+ and 85M+ to match the general Vision Transformer backbones. For object detection/instance segmentation, all models use the framework of Mask-RCNN with the 1x schedule. The results are shown below. It can be found that FAT has great scalability. In addition, our model also demonstrates significant advantages in downstream tasks (object detection/instance segmentation). Compared to recent CMT-S, FAT-B3 has **+3.0**$AP^b$ and **+2.4**$AP^m$.
>
> |Model|Params(M)|FLOPs(G)|Top1-acc(%)|$AP^b$|$AP^m$|
> |--|--|--|--|--|--|
> |Swin-T|29|4.5|81.3|42.2|39.1|
> |UniFormer-S|24|4.2|82.9|45.6|41.6|
> |MOAT-0 [3]|28|5.7|83.3|--|--|
> |CMT-S [4]|25|4.0|83.5|44.6|40.7|
> |FAT-B3|29|4.4|**83.6**|**47.6**|**43.1**|
> |||||||
> |||||||
> |Swin-S|50|8.7|83.0|44.8|40.9|
> |Focal-S|51|9.1|83.5|47.4|42.8|
> |UniFormer-B|50|8.3|83.9|47.4|43.1|
> |MOAT-1 [3]|42|9.1|84.2|--|--|
> |CMT-B [4]|46|9.3|84.5|--|--|
> |FAT-B4|52|9.3|**84.8**|**49.7**|**44.8**|
> |||||||
> |||||||
> |CSwin-B|78 |15.0|84.2|--|--|
> |MOAT-2 [3]|73|17.2|84.7|--|--|
> |CMT-L [4]|75|19.5|84.8|--|--|
> |iFormer-L |87|14.0|84.8|--|--|
> |FAT-B5|88|15.1|**85.2**|--|--|
>
> **Q2:** Efficiency of the method.
>
> **R2:** Thanks. We compare FAT with the recent lightweight Vision Transformer and strong lightweight CNN backbone (EfficientNet, MobileOne, ParC-Net, MobileNetv2). We compare their **inference latency** on GPU/CPU and **inference throughput** on GPU. The CPU is Intel i9Core and the GPU is V100. **Inference throughput** is measured with batch size 64. **Inference latency** is measured with batch size 1. It can be seen that our FAT has the best speed/performance trade-off among lightweight Transformers.
> |Model|Params(M)|FLOPs(G)$\downarrow$|CPU(ms)$\downarrow$|GPU(ms)$\downarrow$|Trp(imgs/s)$\uparrow$|Top1-acc(%)|
> |---|---|---|---|---|---|---|
> |EdgeViT-XXS|4.1|0.6|43.0|14.2|1926|74.4|
> |MobileNetv2_1.4x [6]|6.1|0.6|37.2|11.1|2342|74.7|
> |MobileViT-XS|2.3|1.1|100.2|15.6|1367|74.8|
> |EdgeNext-XS|2.3|0.5|52.3|13.8|1417|75.0|
> |tiny-MOAT-0|3.4|0.8|61.1|14.7|1908|75.5|
> |MobileOne-S1 [5]|4.8|0.8|38.1|5.8|3265|75.9|
> |EfficientNet-B0|5.3|0.4|42.4|14.5|2181|77.1|
> |FAT-B0|4.5|0.7|44.3|14.4|1932|**77.6**|
> |     |   |   |    |    |     |   |
> |     |   |   |    |    |     |   |
> |MobileOne-S2 [5]|7.8|1.3|56.8|6.9|2312|77.4|
> |EdgeViT-XS|6.7|1.1|62.7|15.7|1528|77.5|
> |tiny-MOAT-1|5.1|1.2|80.8|14.8|1506|78.3|
> |MobileViT-S|5.6|2.0|135.6|16.2|898|78.4|
> |ParC-Net-S|5.0|1.7|112.1|15.8|1321|78.6|
> |EfficientNet-B1|7.8|0.7|61.3|18.5|1395|79.2|
> |EdgeNext-S|5.6|1.3|86.4|14.2|1243|79.4|
> |FAT-B1|7.8|1.2|62.6|14.5|1452|**80.1**|
> |     |   |   |    |     |    |   |
> |     |   |   |    |    |     |   |
> |MobileOne-S4 [5]|14.8|3.0|110.2|9.8|1225|79.4|
> |ParC-ResNet50|23.7|4.0|160.0|16.6|1039|79.6|
> |EdgeViT-S|11.1|1.9|95.8|16.2|1049|81.0|
> |tiny-MOAT-2|9.8|2.3|122.1|15.4|1047|81.0|
> |EfficientNet-B3|12.0|1.8|124.2|25.4|624|81.6|
> |FAT-B2|13.5|2.0|93.4|14.6|1064|**81.9**|
>
> **Q3:** Motivation
>
> **R3:** The motivation behind proposing bi-directional interaction in FAT is primarily inspired by existing research on the human brain. In [1], it has been indicated that the human brain has the ability to interpret local information as global shapes. Additionally, in [2], it has been pointed out that the global coordination of local information has significant effects on the processing of the human mid-level visual cortex. These two behaviors align with the examples mentioned in our Introduction, suggesting that the information flows of local to global and global to local should objectively exist in the human brain. Based on these perspectives, we separately model these two types of information flows to obtain modules more similar to the human brain. Since our goal is to design an efficient visual backbone, we did not employ more complex modeling techniques. It has been proven that even with this simple bi-directional interaction pattern, the model can provide significant improvements.
>
> **Q4:** Training stability
>
> **R4:** Thanks. We plot the "epoch-loss" and "epoch-Top1acc" for FAT-B3 in the global response. We invite the reviewer to read it.
>
> **Reference**
>
> [1]Schwarzkopf D. Samuel and Rees Geraint. 2011 Interpreting local visual features as a global shape requires awareness Proc. R. Soc. B.278: 2207–2215. doi/10.1098/rspb.2010.1909
>
> [2]Mannion DJ, Kersten DJ, Olman CA. Regions of mid-level human visual cortex sensitive to the global coherence of local image patches. J Cogn Neurosci. 2014 Aug;26(8):1764-74. doi: 10.1162/jocn_a_00588. Epub 2014 Feb 24. PMID: 24564470; PMCID: PMC4074231.
>
> [3]Chenglin Yang, et al. Moat: alternating mobile convolution and attention brings strong vision models. ICLR, 2023.
>
> [4]Jianyuan Guo, et al. Cmt: Convolutional neural networks meet vision transformers. CVPR, 2022.
>
> [5]Pavan. K, et al. MobileOne: An Improved One millisecond Mobile Backbone. arXiv:2206.04040, 2022.
>
> [6]Sandler. M, et al. Mobilenetv2: Inverted residuals and linear bottlenecks. CVPR, 2018.

---

> > ### Comment · Reviewer_cvP4 · 2023-08-15
> > **Thanks for the rebuttal**
> >
> > I appreciate the additional scaling experiments and the more detailed evaluation of inference efficiency. Most of my concerns have been addressed. While I still have reservations about the motivation, the results presented by the author demonstrate a commendable trade-off between efficiency and the model's performance, especially for downstream tasks. I have revised my scores upward.

---

> > > ### Author Response · Authors · 2023-08-17
> > > **Thanks for the reviewer cvP4's comments**
> > >
> > > Dear reviewer cvP4,
> > >
> > > We want to express our appreciation to the reviewer for the reviewer's time and effort spent reviewing our manuscript. We appreciate the reviewer's recognition of our work and the valuable suggestions provided for its improvement. As for the concerns raised by the reviewer regarding our motivation, we would like to address them further. The experiments conducted in references [1] and [2] provide further evidence for the presence of interaction within the human brain.
> > >
> > > In [1], the authors measured patients' response times after local and global stimuli were presented in the central visual field. Patients with left-hemisphere brain damage showed slower response times in recognizing local stimuli. In comparison, those with right-hemisphere brain damage showed slower response times in recognizing global stimuli, indicating that the left hemisphere specializes in representing local information, whereas the right hemisphere specializes in representing global information.
> > >
> > > In [2], hierarchical stimuli were presented to the human brain, containing both global and local levels nested within each other. Normally functioning individuals are able to perceive both nested levels simultaneously, but left-hemisphere brain-damaged patients faithfully depict the outline of the stimuli without mentioning any local elements. On the other hand, right hemisphere-damaged patients can only draw the local stimuli. Hence, communication between the left and right hemispheres, i.e., the interaction between local and global information, is theoretically necessary for visual recognition.
> > >
> > > Once again, we sincerely appreciate the valuable feedback from the reviewer, which has contributed significantly to the refinement of our work.
> > >
> > > Best regards,
> > >
> > > The authors
> > >
> > > [1]Marvin R. Lamb, Lynn C. Robertson, Robert T. Knight, Attention and interference in the processing of global and local information: Effects of unilateral temporal-parietal junction lesions, Neuropsychologia, Volume 27, Issue 4, 1989, Pages 471-483.
> > >
> > > [2]David Navon, Forest before trees: The precedence of global features in visual perception, Cognitive Psychology, Volume 9, Issue 3, 1977, Pages 353-383.

---

### Official Review · Reviewer_DKrb · 2023-07-06

**Soundness:** 2 fair
**Presentation:** 3 good
**Contribution:** 2 fair
**Rating:** 6
**Confidence:** 5

**Summary:**

This paper presented an efficient vision transformer backbone for several tasks including classification, segmentation and detection. The key idea of this paper is on a new design of considering the interaction between local and global features.  The experiments on imagenet, AD20K, and COCO  have shown that the proposed method is comparable to most existing methods.

**Strengths:**

This paper has clear contributions on the vision transformer design.
1.  The proposed the local-global feature interaction is well motivated.
2.  The proposed methods has been evaluated with comprehensive experiments.
3. This paper is well written and easy to understand.

**Weaknesses:**

There are some weaknesses in this paper:

1. The area of new backbone is growing rapidly.  Although the proposed method shows marginally improvements over the experimented baselines, compared with concurrent work, it becomes not appealing.
2.  The proposed lightvision transformer is very hard to justify as it is not as efficient as mobile vit but also not sure how it can be scaled up.
Thus it is very hard to justify.
3.  The baseline methods are not consistent.  For example Table 1 and Table 2 pick different methods for similar comparison. For example, ParC-Net should be put into 0~5 while the author intentionally put into different comparison setting to make the number look better.



**Questions:**

The experiments are not fair comparison as some of the numbers are cherry picked.  It will be more convincing to compare with Segnext, ParC-Net, MobileOne and mobileVIT clearly.  As far as the reviewer can see, the proposed method is comparable or weaker than them.

In addition, the ablation studies didn't show the bidirectional interaction is useful or not. For example, given single interaction will it be OK?

**Limitations:**

The key limitation of this paper is the proposed method lack of evidence on whether it can be mobile-friendly or scaled up as VIT liked backbones.   Since the backbone is not general enough, it will has very limited usage.

---

> ### Author Rebuttal · Authors · 2023-08-09
>
> ### **On comparison, efficiency, ablation study and scale-up capability**
>
> We thank the reviewer for recognizing the positive aspects of our work. We will address the reviewer's concerns in the following parts.
>
> **Q1:** Comparison in the paper.
>
> **R1:** Thanks. The number of parameters and the computational complexity (i.e., FLOPs) are essential when evaluating a model's size. These two indicators are equally significant. However, for mobile-friendly models, the speed (inference throughput) should also be taken into consideration. In our paper, we follow the above rules when grouping the data, primarily based on the Params and FLOPs, with inference throughput as a secondary factor. For example, ParC-Net-S (5.0M, 1.7GFLOPs) has small Params similar to FAT-B0 (4.5M, 0.7GFLOPs). However, its FLOPs are much higher than those of FAT-B0, B1 (7.8M, 1.2GFLOPs), and close to B2 (13.5M, 2.0GFLOPs). Therefore, considering both the Params and FLOPs, the most reasonable choice is to compare ParC-Net-S with FAT-B1 (5.0M, 1.7GFLOPs v.s. 7.8M, 1.2GFLOPs). So we think we haven't cherry-picked the data. But for a more rigorous comparison, we group the models strictly based on the number of parameters. The updated table can be found in the global response, and we invite the reviewer to read it. It is important to note that due to the high computational complexity and critical parameter count of ParC-Net-S, we still compare it with FAT-B1. The results in global response also show the superiority of FAT.
>
> **Q2:** Comparison with mobilevit on efficiency.
>
> **R2:** Thanks. The results of Table 1 in our paper indicate that our FATs have higher accuracy and inference throughput than the recent mobile architecture, such as MobileViT, EdgeViT, EdgeNext, and more. We invite the reviewer to read the **Q5** for more comparison. Compared to mobilevit, FAT has the advantage in efficiency (FLOPs, inference throughput, and inference latency) and accuracy.
>
> **Q3:** Whether the bidirectional interaction is useful.
>
> **R3:** Thanks. Here we ablate the local-to-global and global-to-local, respectively, to show the effect of bidirectional interaction.    The results are shown in the below table. It can be seen that the bidirectional interaction plays an important role in FAT.
>
>    |Model|Params(M)|FLOPs(G)|Top1-acc(%)|
>    |-|-|-|-|
>    |no interaction|4.5|0.72|76.2|
>    |w/o global to local|4.5|0.72|76.9|
>    |w/o local to global|4.5|0.72|76.8|
>    |bidirectional interaction|4.5|0.72|77.6|
>
>
> **Q4:** How FAT can scale up.
>
> **R4:** Thanks. We scale up our FAT to common backbone sizes (29M, 4.4GFLOPs, 52M, 9.1GFLOPs, and 88M, 15.1GFLOPs). The results are shown below. We use the framework of Mask-RCNN with 1x schedule to conduct the object detection and instance segmentation. It can be seen that FAT has a good ability to scale up. In addition, our model also demonstrates significant advantages in downstream tasks.
>
> |Model|Params(M)|FLOPs(G)|Top1-acc(%)|$AP^b$|$AP^m$|
> |-|-|-|-|-|-|
> |Swin-T|29|4.5|81.3|42.2|39.1|
> |UniFormer-S|24|4.2|82.9|45.6|41.6|
> |MOAT-0 [1]|28|5.7|83.3|--|--|
> |CMT-S [2]|25|4.0|83.5|44.6|40.7|
> |FAT-B3|29|4.4|**83.6**|**47.6**|**43.1**|
> |||||||
> |||||||
> |Swin-S|50|8.7|83.0|44.8|40.9|
> |Focal-S|51|9.1|83.5|47.4|42.8|
> |UniFormer-B|50|8.3|83.9|47.4|43.1|
> |MOAT-1 [1]|42|9.1|84.2|-|-|
> |CMT-B [2]|46|9.3|84.5|-|-|
> |FAT-B4|52|9.3|**84.8**|**49.7**|**44.8**|
> |||||||
> |||||||
> |CSwin-B|78 |15.0|84.2|-|-|
> |MOAT-2 [1]|73|17.2|84.7|-|-|
> |CMT-L [2]|75|19.5|84.8|-|-|
> |iFormer-L |87|14.0|84.8|-|-|
> |FAT-B5|88|15.1|**85.2**|-|-|
>
> **Q5:** Details comparison with other methods (mobilevit, mobileone, Parc-net, SegNext...).
>
> **R5:** Thanks, for lightweight backbones, we compare their **inference latency** on GPU/CPU and **inference throughput** on GPU, CPU is Intel i9Core, GPU is V100. **Inference throughput** is measured with batch size 64. **Inference latency** is measured with batch size 1. It can be seen that our FAT has the best speed/performance trade-off among lightweight Transformers.
>
> |Model|Params(M)|FLOPs(G)$\downarrow$|CPU(ms)$\downarrow$|GPU(ms)$\downarrow$|Trp(imgs/s)$\uparrow$|Top1-acc(%)|
> |-|-|-|-|-|-|-|
> |EdgeViT-XXS|4.1|0.6|43.0|14.2|1926|74.4|
> |MobileViT-XS|2.3|1.1|100.2|15.6|1367|74.8|
> |EdgeNext-XS|2.3|0.5|52.3|13.8|1417|75.0|
> |tiny-MOAT-0|3.4|0.8|61.1|14.7|1908|75.5|
> |MobileOne-S1 [3]|4.8|0.8|38.1|5.8|3265|75.9|
> |EfficientNet-B0|5.3|0.4|42.4|14.5|2181|77.1|
> |FAT-B0|4.5|0.7|44.3|14.4|1932|**77.6**|
> ||||||||
> ||||||||
> |MobileOne-S2 [3]|7.8|1.3|56.8|6.9|2312|77.4|
> |EdgeViT-XS|6.7|1.1|62.7|15.7|1528|77.5|
> |tiny-MOAT-1|5.1|1.2|80.8|14.8|1506|78.3|
> |MobileViT-S|5.6|2.0|135.6|16.2|898|78.4|
> |ParC-Net-S|5.0|1.7|112.1|15.8|1321|78.6|
> |EfficientNet-B1|7.8|0.7|61.3|18.5|1395|79.2|
> |EdgeNext-S|5.6|1.3|86.4|14.2|1243|79.4|
> |FAT-B1|7.8|1.2|62.6|14.5|1452|**80.1**|
> ||||||||
> ||||||||
> |MobileOne-S4 [3]|14.8|3.0|110.2|9.8|1225|79.4|
> |ParC-ResNet50|23.7|4.0|160.0|16.6|1039|79.6|
> |EdgeViT-S|11.1|1.9|95.8|16.2|1049|81.0|
> |tiny-MOAT-2|9.8|2.3|122.1|15.4|1047|81.0|
> |EfficientNet-B3|12.0|1.8|124.2|25.4|624|81.6|
> |FAT-B2|13.5|2.0|93.4|14.6|1064|**81.9**|
>
> As for SegNext, we compare it with our general backbone. It can be seen that FAT is better than SegNext.
>
> |Model|Params(M)|FLOPs(G)|Top1-acc(%)|
> |-|-|-|-|
> |SegNext-T [4]|4.2|0.7|75.9|
> |FAT-B0|4.5|0.7|**77.6**|
> |SegNext-S [4]|14.0|2.2|81.2|
> |FAT-B2|13.5|2.0|**81.9**|
> |SegNext-B [4]|27|4.5|83.0|
> |FAT-B3|29|4.4|**83.6**|
> |SegNext-L [4]|45|8.2|83.9|
> |FAT-B4|52|9.3|**84.8**|
>
> **Reference**
>
> [1]Chenglin Yang, et al. Moat: alternating mobile convolution and attention brings strong vision models. ICLR, 2023.
>
> [2]Jianyuan Guo, et al. Cmt: Convolutional neural networks meet vision transformers. CVPR, 2022.
>
> [3]Pavan. K, et al. MobileOne: An Improved One millisecond Mobile Backbone. arXiv:2206.04040, 2022.
>
> [4]Meng-Hao Guo, et al. SegNeXt: Rethinking Convolutional Attention Design for Semantic Segmentation, NeurIPS, 2022.

---

> > ### Comment · Reviewer_DKrb · 2023-08-17
> > **feedback**
> >
> > Thanks for all the experiments.  The additional results make me more convinced.

---

> > > ### Author Response · Authors · 2023-08-18
> > > **Thanks for the reviewer DKrb's comments**
> > >
> > > Dear reviewer DKrb,
> > >
> > > Thank you for your recognition of our work. We sincerely appreciate the time and effort you have put into reviewing our paper. Your feedback is crucial for improving our work.
> > >
> > > Best regards,
> > >
> > > The authors

---

### Official Review · Reviewer_c9kv · 2023-07-12

**Soundness:** 2 fair
**Presentation:** 3 good
**Contribution:** 2 fair
**Rating:** 7
**Confidence:** 4

**Summary:**

This paper proposes a new family of light-weight vision transformers named FAT, which enhances the local and global feature fusion by a bi-directional interaction between them. Experiments on image classification, object detection, and semantic segmentation are conducted.

**Strengths:**

1. The paper provides a good practice in designing high performance efficient ViTs, which could be a valuable reference for the future work.

2. The results on several benchmark datasets outperform all the compared models.

**Weaknesses:**

1. An incremental work. There exists tons of works that focus on performing local and global features in ViTs (see Sec. 2 of the paper), this paper only proposes a simple way to enhance the feature fusion. Besides, other increments are actually a combination of existing successful practices. For example, the convolutional and non-overlapping downsample module is from PVTv2, the downsample in MHSA have also been widely used (e.g., PVT), CPE is from [1].

2. Actually, a CVPR 2022 paper MixFormer [2] also proposes a bi-directional interaction of convolutional and self-attention branches. What's the pure difference between this method and MixFormer?

3. It is difficult for me to evaluate the efficacy of the proposed bi-directional interactions. The paper only compares the method with simple fusion baselines, while there exist many global-local feature fusion methods, I cannot figure out which one is better according to Table 1, since many increments are added in this method. I suggest the authors to replace the fusion method in FAT with other popular methods such as the interactions in MixFormer and message token in MSG-Transformer [3], for fair comparisons.

---
**References**

[1] Xiangxiang Chu, Zhi Tian, Bo Zhang, Xinlong Wang, and Chunhua Shen. Conditional positional encodings for vision transformers. In ICLR, 2023.

[2] Chen, Q., Wu, Q., Wang, J., Hu, Q., Hu, T., Ding, E., Cheng, J. and Wang, J., 2022. Mixformer: Mixing features across windows and dimensions. In Proceedings of the IEEE/CVF conference on computer vision and pattern recognition (pp. 5249-5259).

[3] Fang, J., Xie, L., Wang, X., Zhang, X., Liu, W. and Tian, Q., 2022. Msg-transformer: Exchanging local spatial information by manipulating messenger tokens. In Proceedings of the IEEE/CVF Conference on Computer Vision and Pattern Recognition (pp. 12063-12072).

**Questions:**

See Weaknesses.

**Limitations:**

Yes.

---

> ### Author Rebuttal · Authors · 2023-08-02
>
> ### **On Novelty, Difference with MixFormer and Efficacy of Bi-directional interaction**
>
> We thank the reviewer for recognizing the positive aspects of our paper, and we will address the reviewer's concerns in the following parts.
>
> **Q1: Novelty.**
>
> **R1:** Thanks. We want to highlight that the key novelty is that we construct a novel module for local-global fusion, which is simple to implement and has negligible costs. Our bidirectional interaction is inspired by the existing research on the human brain [9, 10]. In contrast to all previous feature fusion methods, our bidirectional adaptive interaction stands out in two significant ways:
>
> 1. Bidirectional Property: We model a bidirectional process between the two before fusing local and global features. This results in strengthened local and global features, respectively. In contrast to commonly used linear fusion, our interaction returns two enhanced vectors representing local and global contexts rather than an inseparable vector mixed with both.
> 2. Adaptive Capability: Taking local features as an example, the global weights we use to enhance them are generated by tokens containing global features, making them context-aware and capable of adapting to the input data. Moreover, we calculate their element-wise product directly when fusing the enhanced local and global features. Our context-aware fusion approach is closer to the attention mechanism than linear fusion.
>
> **Q2: Difference with MixFormer [1].**
>
> **R2:** Thanks. We believe that the key differences between our FAT and MixFormer can be summarized in the following three points:
>
> 1. Different motivations: In FAT, the proposed FASA aims to establish channels for information flow from local to global and global to local, allowing local features to perceive global information and refine global features based on local features. On the other hand, MixFormer is motivated by establishing channels for information flow between an image's spatial and channel dimensions. Information flows in MixFormer enable complementary interaction between window attention (with shared parameters in the channel dimension and weak channel representation) and DWConv (with shared parameters in the spatial dimension and weak spatial representation).
>
> 2. Different branches: Both branches of MixFormer perceive local features, while in FASA, we use a branch for local perception and another for global. Additionally, the limitation in MixFormer [1]'s paper mentions that replacing the window attention branch with global attention would lead to a performance decline, contrary to our model's approach, as shown in the table below.
>
>    | Model| Params(M)| FLOPs(G)|Top1-acc(%)|
>    |--|--|--|--|
>    |Conv + Global attention|4.5|0.72|77.6|
>    |Conv + Window attention|4.4|0.72|75.8|
>
>
>
> 3. Different means of fusing information: Compared to FAT, MixFormer introduces more parameters and FLOPs for better representations in the processes of spatial and channel interaction. Comparison can be found in **Q3**. It can be found that FASA gets better performance (+1.1%) with fewer FLOPs (-0.25G )and Params (-1.7M).
>
> **Q3: Efficacy of  bi-directional interaction**
>
> **R3:** Thanks. We compare our FASA with eight baselines, including the module with local-global fusion (MSG in MSG-Transformer [2], Mix Block in MixFormer [1], R2L in RegionViT [3]) and other self-attention mechanisms with good performance (CSWSA in CSwin-Transformer [4], Max-SA in MaxViT [5], WSA/S-WSA in Swin-Transformer [6], SRA in PVT [7] and LSA/GSA in Twins-SVT [8]). In all baselines, we use the same architecture but only modify the token mixer. The results in the below table can demonstrate the superiority of FASA.
>
> | Model|Params(M)|FLOPs(G)|Top1-acc(%)|
> |--|--|--|--|
> | MSG [2]|4.4|0.77|76.3|
> | Mix Block [1]|6.2|0.97|76.5|
> | R2L [3]|4.8|0.76|76.2|
> | CSWSA  [4]|4.3|0.78|76.8|
> | Max-SA [5]|4.3|0.77|75.7|
> | WSA/S-WSA [6]|4.3|0.76|76.1|
> | SRA [7]|4.4|0.71|76.2|
> | LSA/GSA [8]|4.3|0.71|76.6|
> | FASA|4.5|0.72|**77.6**|
>
>
>
> **Reference**
>
> [1] Chen, Q., Wu, Q., Wang, J., Hu, Q., Hu, T., Ding, E., Cheng, J. and Wang, J. Mixformer: Mixing features across windows and dimensions. In CVPR, 2022.
>
> [2Jiemin Fang , Lingxi Xie, Xinggang Wang .et, al. MSG-Transformer: Exchanging Local Spatial Information by Manipulating Messenger Tokens. In CVPR, 2022.
>
> [3]Chun-Fu (Richard) Chen, Rameswar Panda, and Quanfu Fan. RegionViT: Regional-to-Local Attention for Vision Transformers. In ICLR, 2022.
>
> [4]Xiaoyi Dong, Jianmin Bao, Dongdong Chen, et al. Cswin transformer: A general vision transformer backbone with cross-shaped windows. In CVPR, 2022.
>
> [5]Zhengzhong Tu, Hossein Talebi, Han Zhang, et al. Maxvit: Multi-axis vision transformer. In *ECCV*, 2022.
>
> [6]Ze Liu, Yutong Lin, Yue Cao, Han Hu, Yixuan Wei, Zheng Zhang, Stephen Lin, and Baining Guo. Swin transformer: Hierarchical vision transformer using shifted windows. In *ICCV*, 2021.
>
> [7]Wenhai Wang, Enze Xie, Xiang Li, Deng-Ping Fan, Kaitao Song, Ding Liang, Tong Lu, Ping Luo, and Ling Shao. Pyramid vision transformer: A versatile backbone for dense prediction without convolutions. arXiv preprint arXiv:2103.15808, 2021.
>
> [8]Xiangxiang Chu, Zhi Tian, Yuqing Wang, Bo Zhang, Haibing Ren, Xiaolin Wei, Huaxia Xia, and Chunhua Shen. Twins: Revisiting the design of spatial attention in vision transformers. In *NeurIPS*, 2021.
>
> [9]Schwarzkopf D. Samuel and Rees Geraint. 2011 Interpreting local visual features as a global shape requires awareness  Proc.R. Soc. B.278: 2207–2215. doi/10.1098/rspb.2010.1909
>
> [10]Mannion DJ, Kersten DJ, Olman CA. Regions of mid-level human visual cortex sensitive to the global coherence of local image patches. J Cogn Neurosci. 2014 Aug;26(8):1764-74. doi: 10.1162/jocn_a_00588. Epub 2014 Feb 24. PMID: 24564470; PMCID: PMC4074231.

---

> > ### Comment · Reviewer_c9kv · 2023-08-18
> >
> > I appreciate the detailed responses of the authors, and most of my concerns are addressed. After reading the responses and other reviewers' reviews, I am delighted to raise my evaluation to "Accept" and recommend an acceptance of this paper.

---

> > > ### Author Response · Authors · 2023-08-19
> > > **Thanks for the reviewer c9kv's comments**
> > >
> > > Dear reviewer c9kv,
> > >
> > > We are grateful for the reviewer's high evaluation of our work. The suggestions and comments from the reviewer have been incredibly helpful in enhancing our work. Again, We would like to thank the reviewer for investing their time and effort into reviewing our article.
> > >
> > > Best regards,
> > >
> > > The authors

---

### Author Rebuttal · Authors · 2023-08-09

The global response contains two things:
1. Updated version of Table 1 in the paper.
2. Plots to show the training stability.

---

### Comment · Area_Chair_CpVW · 2023-08-14
**Reviewer-author discussion**

Dear Reviewers,

Please take a moment to read the authors' responses. Your insights and feedback are crucial in ensuring a comprehensive evaluation. Thanks.

AC

---

> ### Comment · Area_Chair_CpVW · 2023-08-16
> **Remaining concerns**
>
> Many thanks to the reviewers who have provided feedback to the authors.
>
> @Reviewer c9kv, and @Reviewer DKrb, We kindly request you to take a moment to review the authors' responses and share your opinions or any remaining concerns at your earliest convenience. Your input is greatly appreciated. Thank you.
>
> AC

---

### Decision · Program_Chairs · 2023-09-21

**Decision:**

Accept (poster)

**Comment:**

The reviewers acknowledge the improvements made by the authors in response to their concerns and appreciate the extra experimental results, e.g., the presented trade-off between efficiency and model performance, particularly for downstream tasks. They highlight the innovation of bidirectional interaction between global and local branches as a key strength, underscored by comprehensive ablation studies. The impressive results of FAT-B0 and B1 models in terms of both throughput and accuracy are seen as valuable insights for future research. The introduction of the FAT lightweight ViT structure, its adaptive self-attention mechanism, and fine-grained downsampling strategy are all recognized as meaningful contributions. Overall, the reviewer's reservations have been alleviated, leading to an upward revision of their scores and endorsing the paper's value in the context of low-latency small models.